# G-protein βγ subunits determine grain size through interaction with MADS-domain transcription factors in rice

Qian Liu[1], Ruixi Han[1,2,3], Kun Wu[1,4], Jianqing Zhang[1,2], Yafeng Ye[1], Shuansuo Wang[1], Jianfeng Chen[1,2], Yajun Pan[1], Qi Li[1], Xiaopeng Xu[1,5], Jiawu Zhou[6], Dayun Tao[6], Yuejin Wu[4] & Xiangdong Fu[1,2]

The simultaneous improvement of grain quality and yield of cereal crops is a major challenge for modern agriculture. Here we show that a rice grain yield quantitative trait locus *qLGY3* encodes a MADS-domain transcription factor OsMADS1, which acts as a key downstream effector of G-protein βγ dimers. The presence of an alternatively spliced protein OsMADS1[lgy3] is shown to be associated with formation of long and slender grains, resulting in increases in both grain quality and yield potential of rice. The Gγ subunits GS3 and DEP1 interact directly with the conserved keratin-like domain of MADS transcription factors, function as cofactors to enhance OsMADS1 transcriptional activity and promote the co-operative transactivation of common target genes, thereby regulating grain size and shape. We also demonstrate that combining *OsMADS1[lgy3]* allele with high-yield-associated *dep1-1* and *gs3* alleles represents an effective strategy for simultaneously improving both the productivity and end-use quality of rice.

[1] The State Key Laboratory of Plant Cell and Chromosome Engineering, Institute of Genetics and Developmental Biology, Chinese Academy of Sciences, 100101 Beijing, China. [2] College of Life Sciences, University of Chinese Academy of Sciences, 100049 Beijing, China. [3] Development Center for Science and Technology, Ministry of Agriculture, 100122 Beijing, China. [4] Institute of Technical Biology and Agriculture Engineering, Hefei Institutes of Physical Science, Chinese Academy of Sciences, 230031 Hefei, China. [5] Root Biology Center, South China Agricultural University, 510642 Guangzhou, China. [6] Food Crops Research, Institute Yunnan Academy of Agricultural Sciences, 650200 Kunming, China. Qian Liu and Ruixi Han contributed equally to this work. Correspondence and requests for materials should be addressed to X.F. (email: xdfu@genetics.ac.cn)

Despite the genetic improvement of grain yield delivered by the exploitation of semi-dwarfism and heterosis over the past 50 years[1–3], a substantial increase in grain productivity of the major crops is required to feed a growing world population[4]. The prime breeding target is to increase both grain size and grain number, because they impact both on yield potential and its end-use quality[5]. However, the simultaneous improvement of grain quality and grain yield is a major challenge because of the well-established negative correlation between these two traits[6], all of which is controlled by quantitative trait loci (QTL) and influenced by environmental changes. Rice is a staple food for nearly one-half the world's population, molecular and genetic basis of main yield components (i.e., tiller numbers per plant, grain numbers per panicle, and 1,000-grain weight) have been extensively investigated in the last decade. Recently, several genes have been shown to control grain size and grain number of rice: Gn1a/OsCKX2[7], DENSE AND ERECT PANICLE1 (DEP1)[8–10], GNP1/GA20ox1[11], IPA1/WFP/OsSPL14[12,13], and NPT1/OsO-TUB1[14] regulate panicle architecture and grain number; GS3[15], TGW6[16], GW8/OsSPL16[5,17], GW7/GL7[17,18], GLW7/OsSPL13[19], GS2/GL2/OsGRF4[20–22], GW2[23], and GW5[24,25] regulate grain size and shape. However, the genetic determinants of other QTLs that regulate grain size and number remain poorly understood. Therefore, identification of allelic variations of other genes associated with improvement of grain yield would facilitate the breeding of new high-yielding rice varieties and may be applicable to other crops.

Natural variants of the rice G-protein γ subunits DEP1[8–10] and GS3[15] have been shown to boost grain yield, but typically show only a mediocre quality of grains. The nature of variations associated with higher yield potential is also quite different for each of the two Gγ subunits[26]. The gs3 loss-of-function allele is associated with the formation of a long and slender grain, resulting in an increase in the size of grains, and the reduction of grain number[15]. In contrast, the dep1-1 gain-of-function allele caused an increase in the number of grains per panicle, but produces smaller grains[8,10]. Indeed, the simultaneous improvement of seed size and number is a major challenge for modern agriculture, because of a genetic trade-off between these two traits that producing more seeds would lead to smaller seeds and vice versa[27]. Grain size and number are inherently connected with floral organ identity and growth, which are tightly regulated by various combinations of MADS-domain transcription factors[28]. In rice, homo- and hetero-dimerization and formation of multiple complex and interaction with other proteins (e.g., transcription factors, kinases, and chromatin modifiers) in higher order complexes have been described, but how the members of these interacting partners execute specific target gene expression is not yet fully understood[29]. In addition, changes in multiple environment factors, including biotic and abiotic stresses affect seed size and number. For example, plants grown in resource-limited environments may benefit from producing large seeds of small crops, resulting in offspring seedlings capable of survival and growth under fluctuating environments[30]. However, the molecular mechanisms underlying the interplay between floral organ identity specification and growth capacity in response to environmental changes still remain unknown.

Here we show that a quantitative trait locus LGY3 is synonymous with OsMADS1[29], which encodes a MADS-domain transcription factor. Our results show that a natural variant of OsMADS1[lgy3], an alternatively spliced form of OsMADS1, is associated with the formation of a more slender grain and better appearance quality. We demonstrate that the rice Gβγ subunits act as conserved interacting cofactors with OsMADS1. DEP1 and OsMADS1 share common targets in vivo and their physical interaction is important in driving downstream transcriptional expression profiles and in regulating grain size and shape. Our findings reveal a new molecular framework for the control of variation in seed size and number in response to fluctuating environments. More importantly, haplotype analysis of the OsMADS1 gene reveals that the L-204 haplotype involving truncating splice-site mutation is common within O. nivara accessions and tropical japonica germplasm, but it does not appear to occur within the elite indica and temperate japonica rice varieties, thus combining the OsMADS1[lgy3] allele with the high-yielding dep1-1 and gs3 alleles provides a new strategy for simultaneously improving grain quality and yield potential of rice.

## Results

**Identification of qLGY3.** The japonica rice variety Wuyunjing7 carrying the dep1-1 allele (hereafter WYJ7-dep1-1) produces more, but smaller grains than its near-isogenic line (NIL) WYJ7-DEP1[10] (Fig. 1a). A set of 250 recombinant inbred lines (RILs) were developed from the cross between Wuyunjing7 and the American japonica rice variety L-204, which produces long and slender grains, one line (RIL186) carrying the dep1-1 allele formed grains that were bigger than that formed by the WYJ7-dep1-1 parent (Fig. 1a). A subsequent QTL analysis revealed the presence of a quantitative trait locus qLGY3, which was pleiotropically responsible for long-grain and high-yield traits (Fig. 1b). Genetic analysis of BC$_4$F$_2$ progenies derived from a crossing of RIL186 with the indica rice variety HJX74 (the recurrent parent) suggested that a semi-dominant qlgy3 allele from L-204 was associated with the characteristics of both long-grain and high-yield (Supplementary Fig. 1).

**OsMADS1[lgy3] is associated with long and slender grains.** Positional cloning of qlgy3 was performed by using BC$_2$F$_2$ and BC$_3$F$_2$ populations developed from the cross between R186 and Thai cultivar RD23 (the recurrent parent), and candidate region was narrowed to a ~8.9 Kbp stretch flanked by molecular markers XP22 and XP23 (Fig. 1c). This segment only contains the coding and 3′ untranslated regions of OsMADS1, a gene encoding MADS-domain transcription factor[29]. Sequence analysis indicated that an insertion–deletion polymorphism in the splice site of the intron 7/exon 8 junction was differentiated between L-204 and RD23 (Fig. 1d), resulting in L-204 producing an alternatively spliced protein OsMADS1[lgy3], in which the terminal 37 residues were truncated, and an additional 5 residues were added to its predicted C domain (Fig. 1d). The effect of truncating splice-site mutation was to increase slightly the transcript and protein accumulation, without affecting nuclear localization and its interaction with other MADS-domain proteins[31] (Supplementary Fig. 2).

A NIL NPB-lgy3 was created in the japonica variety Nipponbare background by introgressing a ~274-Kbp segment from L-204 (Supplementary Fig. 3a). The NPB-lgy3 plants formed grains that were longer than those formed by Nipponbare control plants (hereafter NPB-LGY3), but there was little difference with respect to grain width (Supplementary Fig. 3b–d). Furthermore, either length or width of NPB-lgy3 outer epidermal cells was indistinguishable from that of the equivalent cells in NPB-LGY3 (Supplementary Fig. 3e, f). These observations suggested that the lgy3 allele was correlated with the production of more slender grains as results of an increased cell division in the longitudinal direction, but no changes in cell proliferation and elongation in the transverse direction. The grains formed by the transgenic NPB-LGY3 plants expressing the L-204 lgy3 cDNA driven either by its native promoter or by the rice Actin promoter were longer than those formed by non-transgenic control plants, as well as

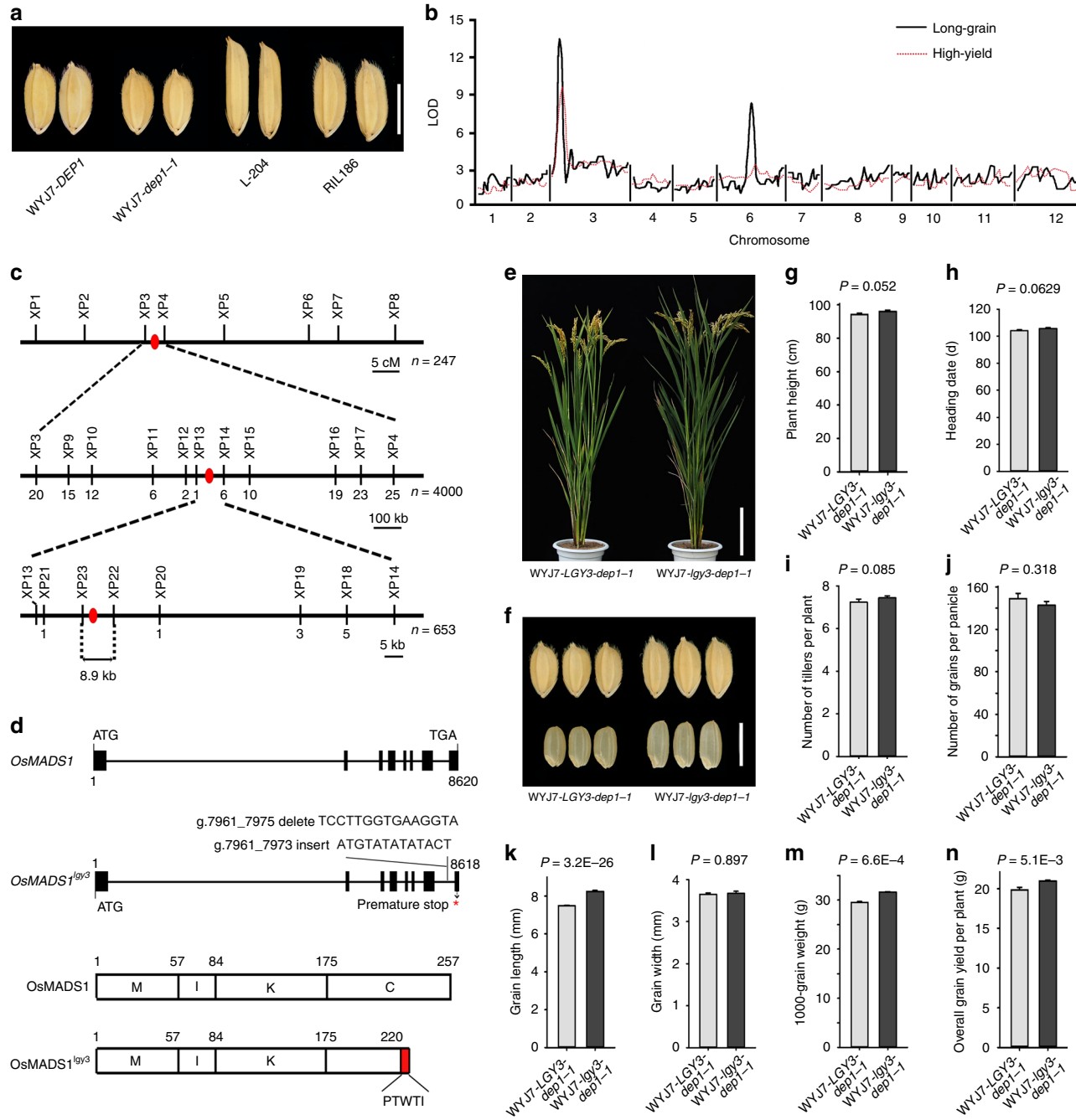

**Fig. 1** The interaction between *DEP1* and *OsMADS1* regulates grain size and yield potential of rice. **a** Grain morphology. Scale bar: 5 mm. **b** QTL mapping for grain length and grain yield. **c** The positional cloning of *qLGY3*. The candidate region was mapped to a ~ 8.9 Kbp genomic DNA region lying between markers XP22 and XP23. The numbers below the line indicate the number of recombinants recovered between *qLGY3* and markers. **d** Allelic variations of *OsMADS1* between RD23 and L-204. In the schematic illustration of OsMADS1 functional domains, M represents the MADS domain, I represents the intervening domain, K represents the keratin-like domain, and C represents the C-terminal domain. **e** The gross morphology of NIL plants. Scale bar: 15 cm. **f** Grain size and shape. Scale bar: 5 mm. **g**–**n** A field-based comparison of the WYJ7-*LGY3-dep1-1* and WYJ7-*lgy3-dep1-1* plants: **g** Plant height; **h** Heading date; **i** The number of tillers per plant; **j** The number of grains per panicle; **k** Grain length; **l** Grain width; **m** 1,000-grain weight; and **n** The overall grain yield per plant. All phenotypic data were measured from the paddy-grown NIL plants grown under normal cultivation conditions. Data shown as mean ± s.e.m. (*n* = 288). Student's *t*-test was used to generate the *P* values

those produced by transgenic NPB-*LGY3* plants in which *OsMADS1* had been knocked down by RNAi (Supplementary Fig. 3g, h). There was little difference in the length of grains formed by over 10 independent transgenic rice plants expressing the Nipponbare *LGY3* cDNA driven by its native promoter, whereas constitutive expression of the Nipponbare *LGY3* cDNA driven by the rice *Actin* promoter resulted in abnormalities in the lemma and palea development (Supplementary Fig. 3h), a phenotype which was indistinguishable from that formed by *Ubi::OsMADS1* plants as described previously[32]. Thus, the product of *OsMADS1*[lgy3] appears to be a dominant negative regulator of OsMADS1 function.

**Combining *lgy3* and *dep1-1* alleles improves quality and yield.** Haplotype analysis at the *OsMADS1* locus revealed that the L-204 haplotype involving truncating splice-site mutation is common within *O. nivara* accessions and tropical *japonica* germplasm, but it does not appear to occur within the elite *indica* and temperate *japonica* rice varieties (Supplementary Data 1), indicating that the *lgy3* allele has not been exploited in high-yielding rice breeding programs. The *dep1-1* allele on the other hand has been widely used to develop high-yielding rice varieties by Chinese breeders[8,10], therefore, it was of interest to characterize the effect on grain yield of combining elite alleles at the *LGY3* and *DEP1* loci. The two NILs plants, WYJ7-*lgy3-dep1-1* and WYJ7-*LGY3-dep1-1*, did not differ from one another with respect to heading date, plant height, tiller numbers per plant, and grain numbers per panicle (Fig. 1e, g–j), but WYJ7-*lgy3-dep1-1* plants produced longer and heavier grains than those formed by WYJ7-*LGY3-dep1-1* plants (Fig. 1f, k–m). Over three successive years of field trialling, WYJ7-*lgy3-dep1-1* out-yielded WYJ7-*LGY3-dep1-1* by an average of 10.4% (Fig. 1n). Meanwhile, WYJ7-*lgy3-dep1-1* plants produced much better quality in terms of grain length-to-width ratio and grain chalkiness (Fig. 2 and Supplementary Data 2). Thus, pyramiding of the *lgy3* and *dep1-1* alleles provides a promising strategy for simultaneously improving rice yield and grain quality above what is currently achievable.

**DEP1 interacts with MADS-domain transcription factors.** In rice, *OsMADS1* is one of E-class MADS-box genes involved in the ABCDE model of flower development[33,34]. To uncover the molecular mechanisms underlying genetic interaction between the *DEP1* and *OsMADS1* genes, we performed a yeast two-hybrid screen to identify DEP1-interacting proteins, and found that both DEP1 and truncated dep1-1 proteins interacted with OsMADS1 (Supplementary Fig. 4a). The DEP1–OsMADS1 interactions in

planta were confirmed by the split firefly luciferase complementation (SFLC) in tobacco leaf epidermal cells and co-immunoprecipitation assays (Fig. 3a, b). We also found that OsMADS1[lgy3] could interact with both DEP1 and dep1-1 proteins (Fig. 3a), implying that it was unlikely that the C domain of OsMADS1 was critical for its interaction with DEP1.

We next constructed various domain-based deletions of OsMADS1 for SFLC assays, and found that both I and K domains were necessary and sufficient for the interaction with DEP1 (Fig. 3c). Previous studies have shown that the K domain is required for MADS-domain protein function and is highly conserved across many plant taxa[35], which is consistent with the findings that DEP1 could interact with other MADS-domain proteins (Supplementary Fig. 4b). In addition, we constructed various domain-based deletions of DEP1[10], and further SFLC assays demonstrated that the vWFC domain of DEP1 was required for the DEP1–OsMADS1 interaction (Supplementary Fig. 5). The transient transcriptional activity assays showed that C domain-dependent OsMADS1 transactivation activity was substantially enhanced when both OsMADS1 and DEP1 were combined (Fig. 3d). Although transactivation activity of either OsMADS1$^{\Delta K}$ (lacking the K domain) or OsMADS1$^{\Delta I+\Delta K}$ (lacking both I and K domains) was stronger than that of OsMADS1, its activity was not promoted, but was retained when OsMADS1$^{\Delta K}$ (or OsMADS1$^{\Delta I+\Delta K}$) and DEP1 were combined (Fig. 3d). These results suggest that DEP1 physically interacts with transcription factor OsMADS1, and the DEP1–OsMADS1 interaction promotes the transactivation activity of OsMADS1.

**OsMADS1 and DEP1 co-regulate common target genes.** The constitutive knockdown of *OsMADS1* in both WYJ7-*DEP1* and WYJ7-*dep1-1* plants resulted in a phenotype similar to that of *osmads1* mutant[29,31], which produced an elongated palea and lemma (Supplementary Fig. 6). These genetic results suggest that *DEP1*-mediated regulation of grain size and shape is dependent upon *OsMADS1* function. To identify common target genes of the *DEP1-OsMADS1* regulatory module, we next compared the genome-wide transcriptional profiles in the developing panicles and spikelet hulls of the NILs WYJ7-*LGY3-DEP1*, WYJ7-*LGY3-dep1-1*, WYJ7-*lgy3-DEP1,* and WYJ7-*lgy3-dep1-1* plants using RNA-seq. Comparisons of RNA-seq and ChIP-seq data[36] revealed a total of 451 genes, which were cooperatively regulated by the *DEP1-OsMADS1* regulatory module (Fig. 4a and Supplementary Data 3). Among these were certain AP2/ERF and MADS-box family members, along with genes encoding proteins involved in the biosynthesis of, transport of, and response to auxin (Supplementary Data 3). Quantitative RT-PCR assays confirmed that the GARP family genes *OsKANADI2* and *OsKANADI4*, the MADS-box family genes *OsMADS34* and *OsMADS55*, the auxin efflux carrier *OsPIN1a*, and the auxin response factor genes *OsARF9* and *OsARF14* were all more strongly transcribed in WYJ7-*LGY3-dep1-1* plants than in either WYJ7-*LGY3-DEP1* or WYJ7-*lgy3-dep1-1* plants (Fig. 4b). These results indicate that *dep1-1* and *OsMADS1* cooperatively regulate a common set of target genes during development of spikelets in rice. Further ChIP-PCR and EMSA assays revealed that both OsMADS1 and OsMADS1[lgy3] were able to bind the promoter regions of *OsMADS55*, *OsKANADI4*, *OsPIN1a*, and *OsARF9*, and that the DNA-binding affinity was unaffected by variations in the C domain sequence (Fig. 4c, d).

Moreover, ChIP-PCR assays demonstrated that the in vivo association of OsMADS1 with a same promoter region of target genes examined was unaffected by the presence or absence of dep1-1 (Fig. 4e), indicating that the effect of the DEP1–OsMADS1 interaction on target gene expression was not

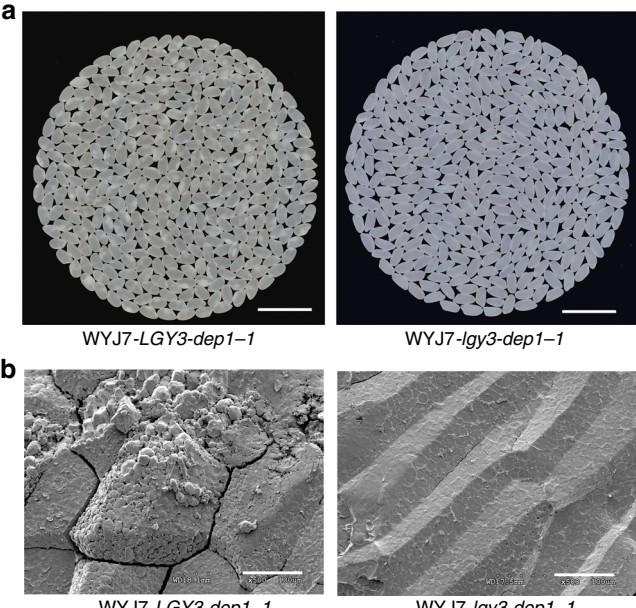

**Fig. 2** Introduction of the *lgy3* allele into the high-yielding variety results in the improvement of rice grain quality. **a** Comparisons of grain chalkiness and endosperm transparency between the WYJ7-*LGY3-dep1-1* and WYJ7-*lgy3-dep1-1* plants. Scale bar: 15 mm. **b** Scanning electron microscopy images of the transverse sections of starch granule from the WYJ7-*LGY3-dep1-1* and WYJ7-*lgy3-dep1-1* plants. Scale bar: 50 μm. The endosperm of the NIL plants carrying the *lgy3* allele comprised largely sharp edged, compactly arranged polygonal starch granules

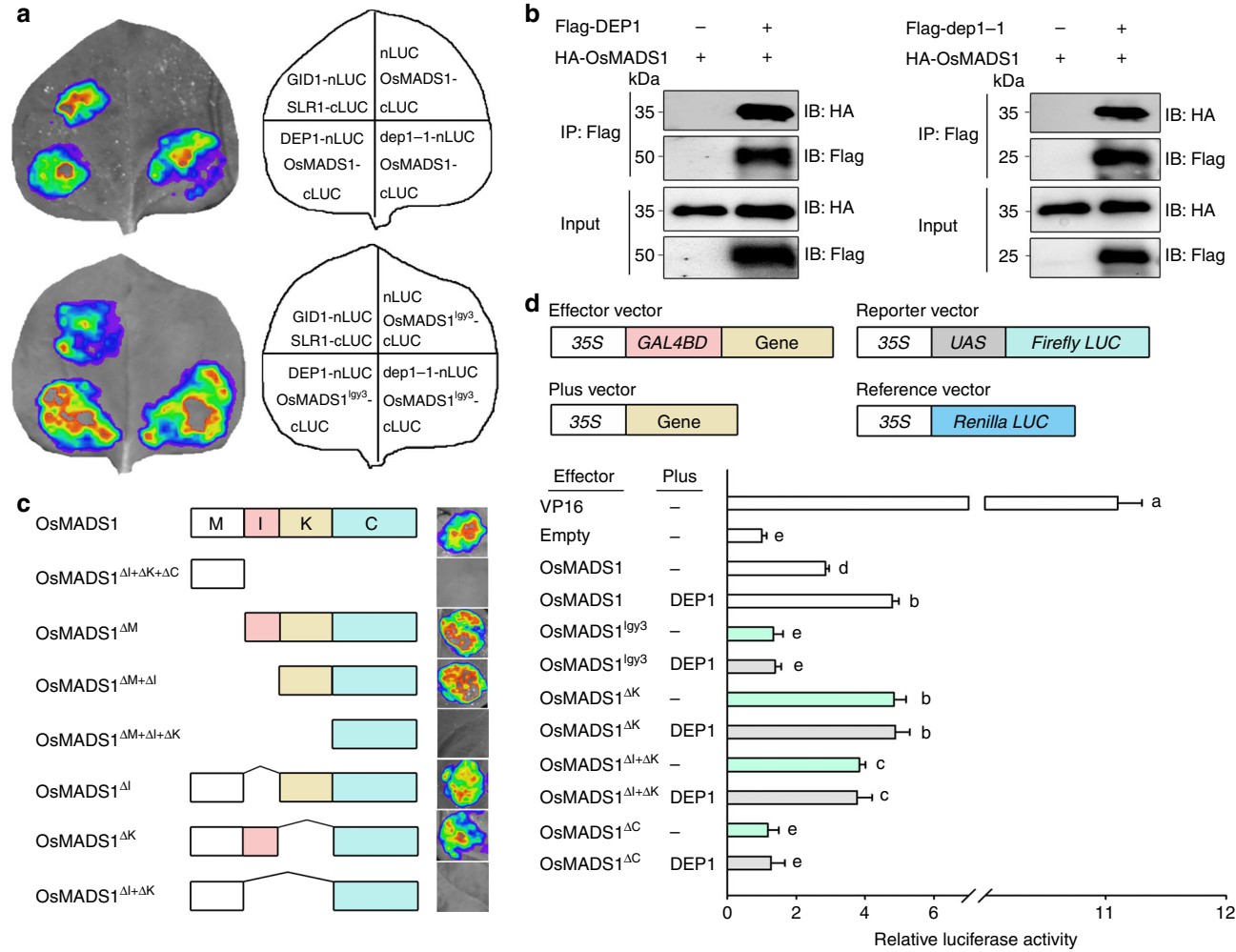

**Fig. 3** The physical interaction between DEP1 and OsMADS1. **a** SFLC assays. nLUC-tagged DEP1 (or dep1-1) was co-transformed into tobacco leaves along with either the cLUC-targeted OsMADS1 or cLUC-targeted OsMADS1$^{lgy3}$. The interaction of the rice GA receptor GID1 and the rice DELLA protein SLR1 were used as positive control. **b** Co-immunoprecipitation assays of Flag-DEP1 and HA-OsMADS1, and Flag-dep1-1 and HA-OsMADS1. IB Immunoblot, IP immunoprecipitation. **c** The image shows a SFLC assay in which nLUC-tagged deleted and non-deleted versions of OsMADS1 were co-transformed into tobacco leaves along with cLUC-DEP1. **d** The effect of the OsMADS1-DEP1 interaction on OsMADS1-induced transactivation activity. Deleted and non-deleted versions of OsMADS1 were fused to the GAL4-binding domain (GAL4BD). The relative activity of firefly luciferase (LUC) under control of the GAL4-binding element *UAS* was measured. Renilla luciferase (REN) activity was used as reference and VP16 as the positive control. Data shown as mean ± s.e.m. (n = 6). Statistical analyses were performed by Duncan's multiple range tests. The presence of the same lowercase letter denotes a non-significant difference between means (P > 0.05)

triggered by the resulting changes in the DNA-binding affinity of OsMADS1. Rice OsARF9 is homologous to *A. thaliana* ARF2 and *S. lycopersicum* SlARF9, which negatively control, respectively, seed size in *A. thaliana*[37] and fruit weight in tomato[38]. Transient expression assays in rice protoplasts showed that LUC activity driven by the *OsARF9* promoter was moderately induced by OsMADS1 on its own, but was substantially enhanced when both OsMADS1 and DEP1 were combined (Fig. 4f), which is well consistent with the experimental results that *OsMADS1* and *DEP1* acted as the repressors in regulating grain size (Fig. 1a, f). Furthermore, LUC activity was moderately induced by OsMADS1$^{ΔK}$ on its own, but not enhanced when both DEP1 and OsMADS1$^{ΔI+ΔK}$ were combined (Fig. 4f). In addition, OsMADS1-induced LUC activities were promoted when assays were performed in the presence of either DEP1 or dep1-1, whereas mutations in the C domain of OsMADS1 abolished the DEP1-promoted transactivation activity (Fig. 4g). Taken together, these results indicate that DEP1 acts as a functional OsMADS1

cofactor in controlling grain size through regulation of the common target genes.

**OsMADS1 is a key downstream effector of G-protein βγ dimers.** Heterotrimeric GTP-binding proteins (G proteins) are signal transduction components that mediate multiple intracellular responses to external stimuli in diverse eukaryotic organisms. The rice genome encodes one Gα subunit (RGA1), one Gβ subunit (RGB1), two canonical Gγ subunits (RGG1 and RGG2), and three non-canonical Gγ subunits (GS3, DEP1, and OsGGC2)[39]. It is well-known that, following dissociation of Gα subunit, Gβγ dimers act as a functional unit to activate a large number of its own effectors[40]. Previous studies have shown that RGB1, RGG1, and RGG2 were datable in the plasma membrane[41]. However, recent studies suggested potential nuclear localization or translocation of the Gβγ dimer in mammalian cells, which co-localized with the AP-1 transcription factor and recruited

histone deacetylases to inhibit AP-1 transcriptional activity[42,43]. To investigate whether transcription factor OsMADS1 acts as a downstream effector of Gβγ subunits in rice, we performed the analysis of protein–protein interactions in planta. Although the canonical G-protein γ subunits RGG1 and RGG2 do not contain vWFC domain, SFLC assays showed that all of the rice Gγ subunits interacted with OsMADS1 (Fig. 5a). In addition, the rice Gβ subunit RGB1, but not the rice Gα subunit RGA1, was associated

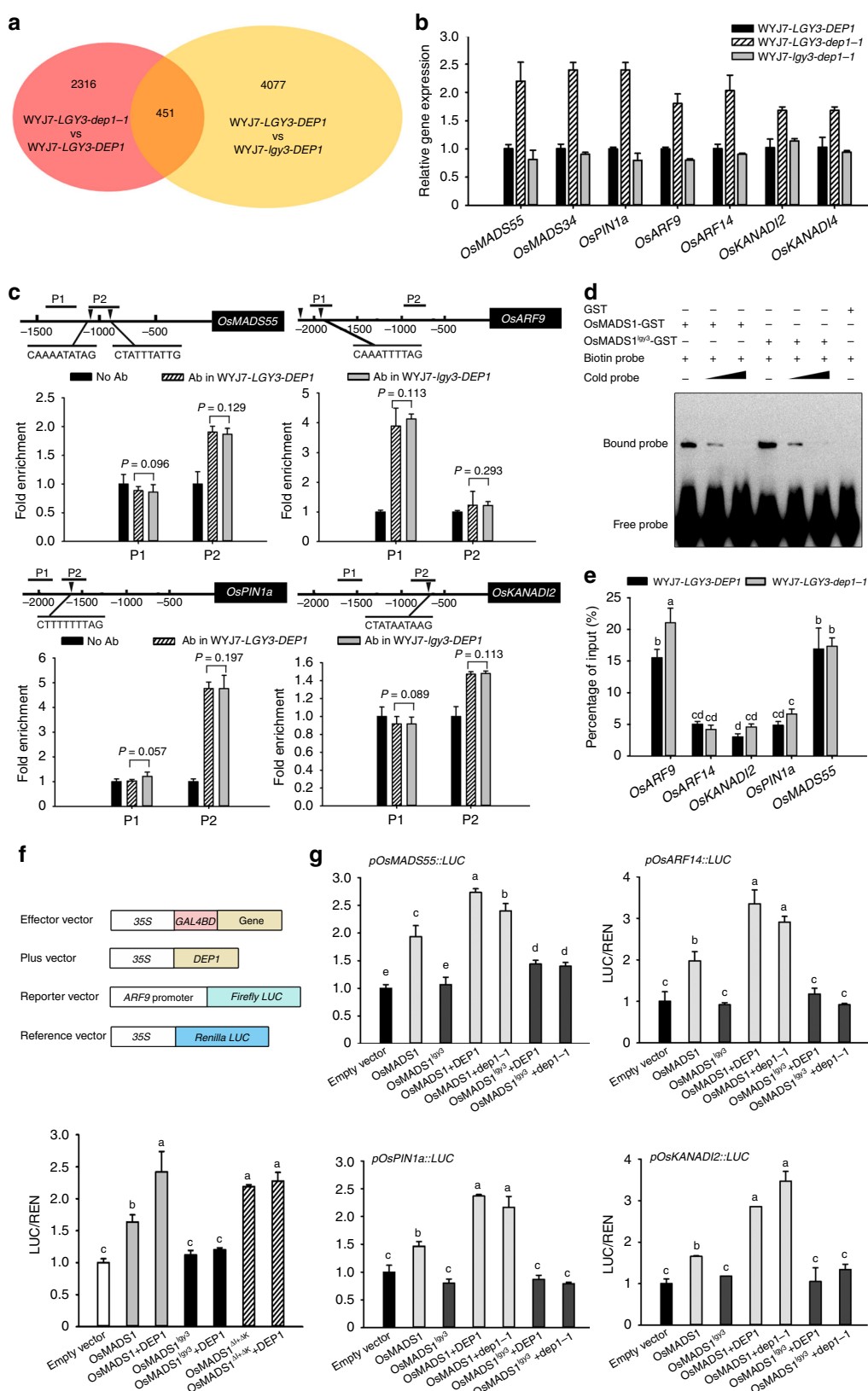

with OsMADS1 (Fig. 5a). Interestingly, the nuclear-localized RGB1-GFP fusion proteins were datable in both the plasma membrane and the nucleus of root cells in transgenic rice plants expressing *RGB1-GFP* under the control either of its native promoter (Supplementary Fig. 7a) or of the cauliflower mosaic virus (CaMV) 35S promoter[10]. Further transient expression assays showed that dep1-1-GFP and GS3-GFP fusion proteins were datable in both the plasma membrane and the nucleus (Supplementary Fig. 7b). To investigate whether the nuclear-localized dep1-1 protein was involved in controlling plant height and grain size, we generated transgenic rice plants carrying either *pActin::HA-NLS-dep1-1* or *pDEP1::HA-dep1-1* construct. We found that the nuclear-localized HA-NLS-dep1-1 fusion protein was associated with the formation of a semi-dwarf phenotype (Supplementary Fig. 7c), which was similar to that of the transgenic rice plants expressing *dep1-1* under control of either the constitutive promoter[10] or its native promoter (Supplementary Fig. 7c). These results suggest that the nuclear translocation of DEP1 is involved in G-protein signaling in rice.

The transgenic rice plants carrying the *pActin::GS3-GFP* construct also displayed a semi-dwarf phenotype with the reduced size of grains (Supplementary Fig. 7d), the phenotypes that were indistinguishable from that of the transgenic plants carrying the *pDEP1::dep1-1* construct[10]. We also showed that the GS3-GFP fusion protein was detectable in the plasma membrane and the nucleus of rice root cells (Supplementary Fig. 7e, f). In addition, the transgenic rice plants overexpressing either *RGB1*, *RGG1*, or *RGG2* formed grains that were much smaller than those formed by non-transgenic control plants, a phenotype reminiscent of that of *dep1-1* overexpressor[10] (Fig. 5b). In addition, introduction of the *lgy3* allele into the transgenic rice plants overexpressing either *RGB1* or *HA-RGG1* substantially enhanced grain length and formed more slender grains (Fig. 5b). Further protoplast transient expression assays showed that OsMADS1-induced LUC activities of the common target genes examined were substantially enhanced when GS3 and OsMADS1 were combined (Fig. 5c), representing a response similar to that regulated by the DEP1−OsMADS1 interaction (Fig. 4g). Taken together, these results reveal that both DEP1 and GS3 appear to be cofactors of OsMADS1 in controlling the grain size through regulation of common target genes.

### Combining *OsMADS1^{lgy3}* with *gs3* enhances yield and quality.
The *gs3* allele has been used widely in *indica* rice breeding programs[5,15]. The effect of allelic combinations on grain yield and quality was explored by generating NILs carrying allelic combinations of *qGS3* and *qLGY3* loci in the *indica* variety RD23

(hereafter RD23-*LGY3-gs3*) (Fig. 5d, e). The two NILs RD23-*LGY3-gs3* and RD23-*lgy3-gs3* did not differ from one another with respect to either heading date, plant height, or tiller number (Fig. 5f–h), although the number of grains per panicle was slightly reduced (Fig. 5i). Both RD23-*LGY3-gs3* and RD23-*lgy3-GS3* plants formed longer grains than those formed by RD23-*LGY3-GS3* plants (Fig. 5e), while RD23-*lgy3-gs3* plants formed substantially longer and heavier grains than those formed by either RD23-*LGY3-gs3* or RD23-*lgy3-GS3* plants (Fig. 5e, j, k). Over three successive years of field trials, RD23-*lgy3-gs3* proved to be on average ~10.9% more productive than RD23-*LGY3-gs3* (Fig. 5l). In addition, the introduction of the *lgy3* allele into elite variety RD23 also improved grain appearance quality (Supplementary Figs. 8a, 9a and Supplementary Data 2).

Because the mode of inheritance of the *OsMADS1^{lgy3}* mutation was semi-dominant (Supplementary Fig. 1), we next investigated whether the *lgy3* allele was able to simultaneously improve grain quality and yield in *indica* hybrid rice. The two-line hybrid combinations[44] were developed by crossing a photothermosensitive genic male sterile PA64S either with a restorer line 9311 (which carries the *LGY3* and *gs3* alleles, hereafter 9311-*LGY3-gs3*) or with a 9311 NIL plants (which carries the *lgy3* and *gs3* alleles, hereafter 9311-*lgy3-gs3*). The effects of the *lgy3* allele were to enhance both grain length and weight (Fig. 5m–o), to increase grain yield by a mean of 7.1% (Fig. 4p), and to substantially improve the grain quality (Supplementary Figs. 8b, 9b and Supplementary Data 2). Thus, combining *gs3* with *lgy3* may be an efficient method for breeding higher-quality and higher-yielding rice above what is currently achievable.

### Discussion
Heterotrimeric G-protein signaling is an evolutionarily conserved mechanism found in all eukaryotes. In humans, the heterotrimeric G-protein complexes comprising 23 Gα, 5 Gβ, and 12 Gγ subunits are important signal transducers that mediate intracellular responses to a broad range of external stimuli. Gα subunit is a molecular switch for bimodal activities with a guanosine diphosphate (GDP)-bound "off" mode and a GTP-bound "on" mode[39]. In the absence of ligands, Gα subunit binds to GDP and forms an inactive heterotrimer with Gβγ dimer. In the presence of ligands, G protein-coupled receptors (GPCRs) are activated and then undergo a conformational change, which in turn promote the exchange of GDP for GTP on Gα subunit, thereby triggering specific downstream effectors of both activated Gα and Gβγ subunits[39]. In plants, the G-protein complex is also composed of Gα, Gβ, and Gγ subunits, integrating multiple environmental and hormonal signals, and plays the important roles in the regulation

---

**Fig. 4** DEP1 and OsMADS1 cooperatively regulate a common set of target genes. **a** The number and overlap of DEP1 and OsMADS1 targets. RNA-seq experiments were performed based on template extracted from young NIL panicles. **b** Transcript abundances of the common target genes examined in the indicated NILs plants. The relative abundances in NIL-*LGY3-DEP1* plants was set to be one. Data shown as mean ± s.e.m. ($n = 6$). **c** ChIP assays of the WYJ7-*LGY3-DEP1* and WYJ7-*lgy3-DEP1* plants. The DNA fragments either containing or non-containing CArG-box motifs (arrowed) in the promoter regions of the targets *OsMADS55*, *OsKANADI42*, *OsPIN1a*, and *OsARF9* were used. Data shown as mean ± s.e.m. ($n = 6$). A Student's *t*-test was used to generate the *P* values. **d** EMSA assays. Competition for either OsMADS1-GST or OsMADS1^{lgy3}-GST binding was performed with 10× and 50× unlabeled probes containing the CArG-box motifs, respectively. **e** ChIP analysis of the OsMADS1 interaction with the promoter regions of *OsMADS55*, *OsKANADI2*, *OsPIN1a*, *OsARF9*, or *OsARF14* with amplicon sequences shown in Fig. 3c. Data shown as mean ± s.e.m. ($n = 3$). Statistical analyses were performed by Duncan's multiple range tests. **f** The effect of deleting the K domain on the DEP1-dependent promotion of OsMADS1-induced transactivation activity of the *OsARF9* promoter. The ratio of LUC to REN activity was monitored in rice protoplasts co-transfected with various effector and reporter constructs. The LUC/REN activity obtained from a co-transfection with an empty effector construct and reporter construct was set to be one. Data shown as mean ± s.e.m. ($n = 6$). **g** The DEP1-OsMADS1 interaction promotes the *OsMADS1*-induced target gene transactivation activity. The LUC/REN activity obtained from a co-transfection with an empty effector construct and indicated reporter constructs was set to be one. Data shown as mean ± s.e.m. ($n = 6$). Statistical analyses were performed by Duncan's multiple range tests. The presence of the same lowercase letter denotes a non-significant difference between means ($P > 0.05$)

of plant growth, metabolism, and adaption in relation to biotic and abiotic stress conditions[10,45–50].

In humans, the functional specificity of G-protein signaling is possible through over 1,380 unique combinations of heterotrimeric G proteins[39]. Conversely, most plants have only one Gα, one Gβ, and several Gγ subunits[51]; hence, rising questions regarding how the simple heterotrimer variants controls the various intercellular processes in plants. Unlike human GPCR-

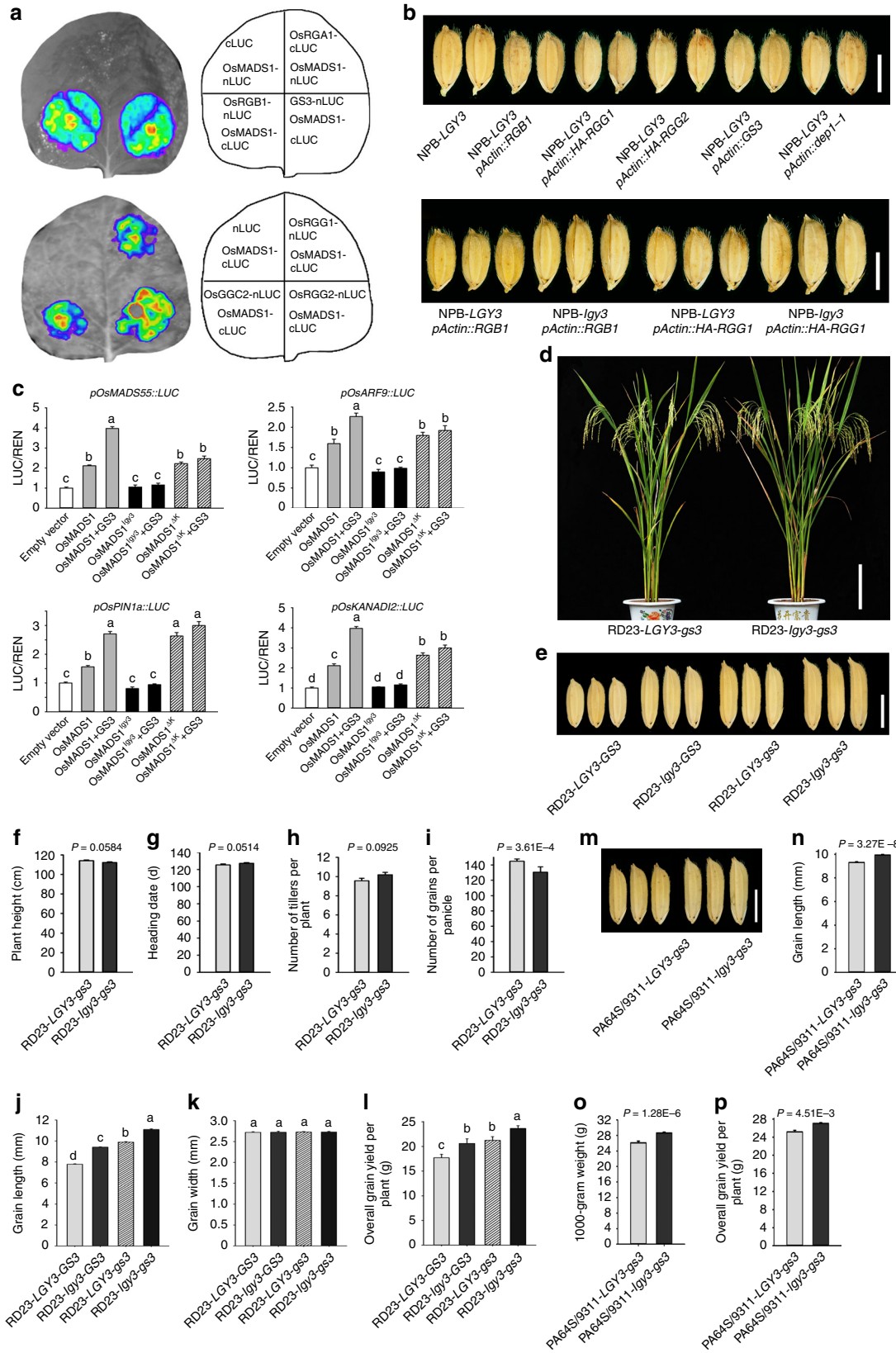

dependent G-protein activation–deactivation cycle, the plant Gα subunit can be self-activation, it was proposed that the acceleration of GTP hydrolysis maybe more important for G-protein activation–deactivation cycle[39]. Recent studies showed that Nod factor receptors (NFR1/NFR5) interacted with soybean Gα and RGS proteins, which in turn promoted the formation of the inactive heterotrimer, thereby regulating the nodule formation and development[52]. It has been shown that maize FEA2 (FASCIATED EAR2) protein, an *Arabidopsis* CLV2 homolog, interacted with maize Gα subunit CT2 (COMPACT PLANT2), resulting in the control of stem cell homeostasis in shoot apical meristems[53]. These results revealed that the plant heterotrimeric G-protein signaling specificity is possible through the formation of multiple complexes between G proteins and receptor-like kinases, and these interacting partners may be essential to sense ligands and distinguish extracellular stimuli, and may also transduce the signals into the cell and finally induce intercellular responses.

Following stimulation of GPCRs at the cell surface, the activated Gα and detached Gβγ dimers classically act as two functional modules[40], and consequently trigger specific downstream effectors and intercellular responses. However, the signaling mechanisms downstream of the heterotrimeric G protein-mediated intercellular responses remain poorly understood. The role of Gβγ dimers was initially proposed to be prevention of spontaneous activation of the Gα subunit, but systematic screens for the protein–protein interactions with Gβ and Gγ subunits identified several interacting partners that were localized at different sub-cellular sites distinct from the plasma membrane[42,43]. Recent studies demonstrated that Gβγ dimer was able to co-localize with transcription factor AP-1 in the nucleus, and recruited histone deacetylases to inhibit activity of AP-1 transcription[42,43]. Here, we showed that the GS3-GFP fusion protein was able to be detected in the nucleus of protoplasts isolated form leaf sheath of 10-day-old seedlings of the transgenic rice plants (Supplementary Fig. 7), consistent with the pervious observations that both RGB1-GFP and dep1-1-GFP fusion proteins were detectable in the nucleus[10]. The outcome suggests potential nuclear localization or translocation of the Gβγ dimers found in both animals and plants.

In this study, we identified a rice quantitative trait locus *LGY3*, which is synonymous with *OsMADS1*, a gene encoding a MADS-domain transcription factor[29]. In the further experiments, we demonstrated that the *lgy3* allele is a dominant negative mutation causing an alternatively spliced protein OsMADS1[lgy3], in which the terminal 37 residues were truncated and an additional 5 residues were added to its predicted C domain of OsMADS1 (Fig. 1d). The effect of the *lgy3* allele is to promote cell proliferation in longitudinal direction during the spikelet development in rice (Supplementary Fig. 3b–d), resulting in the formation of a more slender grain and better appearance quality,

which also applies in combing with the high-yielding *dep1-1* allele (Fig. 1f, Supplementary Data 2). Indeed, Both RNA-seq and ChIP-seq analysis demonstrated that OsMADS1 and DEP1 share common target genes, such as the genes involved in auxin biosynthetic pathway and the auxin-signaling related genes (Fig. 4a, b and Supplementary Data 3), which is in accordance with the function of dep1-1 in regulating the activity of shoot apical meristems and panicle branching[8,10]. More importantly, we have found that DEP1, as well as truncated dep1-1, interacted directly with OsMADS1 (Fig. 3a, b). Both DEP1 and dep1-1 are positive regulators of OsMADS1-mediated transcription, but this interaction is sufficient, but not necessary for enhancement of the transcriptional activity of OsMADS1 (Fig. 4f, g). SFLC assays also showed that rice Gβ subunit RGB1 could interact directly with OsMADS1 (Fig. 5a), which is consistent with the note that potential nuclear localization or translocation of animal Gβγ dimers[42,43]. Our findings suggest that rice Gβγ dimers can be co-localized (or translocated) into the nucleus and interacted directly with MADS-domain transcription factors (i.e., OsMADS1), thereby regulating downstream target gene expression.

It is known that all set of the MADS-domain proteins contain a DNA-binding domain named MADS domain (M domain) that is located at the N terminal[54]. MIKC-type (also known as type II) MADS-domain proteins contain three additional domains, including an Intervening domain (I domain), Keratin-like domain (K domain), and C-terminal domain (C domain). The K domain, as the most highly conserved and characteristic one, together with I domain are responsible for protein interaction and dimerization[55]. Our results showed that both I domain and K domain were indispensable for the interaction with the Gγ subunit DEP1 (Fig. 3c). The transcriptional activities of OsMADS1 were enhanced when OsMADS1 was combined with either DEP1 or dep1-1 (Fig. 3d), suggesting that Gβγ subunits interact with MIKC-type MADS proteins and led to the formation of multiple proteins complexes and consequent increases in transcription activity. The MADS-domain transcription factors have been shown to play the important roles in the control of flowering time, floral organ identity, and growth in response to changes in the external environments (e.g., temperature)[56,57]. Given the evolutionary conservation of both G proteins and MADS-box gene families, our discovery that the interaction between Gβγ subunits and MADS-domain proteins adds a further dimension to current knowledge about cross-talk between the heterotrimeric G-protein signaling and MADS-domain proteins in the regulation of plant growth and development in response to environmental changes.

It has long been believed that the plants that produce more seeds would produce smaller seeds and the plants that produce larger seeds would produce fewer due to genetic trade-offs between these two traits, which are controlled by QTL and influenced by multiple environmental changes. Previous

**Fig. 5** The Gβγ dimer is a functional OsMADS1 cofactor in regulating grain size. **a** SFLC assays. nLUC-tagged either RGA1, RGB1, RGG1, RGG2, GS3, or OsGGC2 was co-transformed into tobacco leaves along with cLUC-targeted OsMADS1. **b** Grain size and shape. Scale bar: 5 mm. **c** Effect of the GS3–OsMADS1 interaction on OsMADS1-induced transactivation activity. The LUC/REN activity obtained from a co-transfection with an empty effector construct and indicated reporter constructs was set to be one. Data shown as mean ± s.e.m. (*n* = 6). Statistical analyses were performed by Duncan's multiple range tests. **d–l** A field-based comparison of RD23-LGY3-gs3 and RD23-lgy3-gs3 plants: **d** Plant morphology. Scale bar: 20 cm; **e** Grain size and shape. Scale bar: 5 mm; **f** Plant height; **g** Heading date; **h** The number of tillers per plant; **i** The number of grains per panicle; **j** Grain length; **k** Grain width; and **l** The overall grain yield per plant. A Student's *t*-test was used to generate the *P* values (**f–i**), and Duncan's multiple range tests were performed to generate the *P* values (**j–k**). **m–p** A field trial of *indica* hybrid rice: **m** Grain size and shape. Scale bar: 5 mm; **n** Grain length; **o** 1,000-grain weight; and **p** The overall grain yield per plant. All phenotypic data were measured from the paddy-grown plants under normal cultivation conditions. Data shown as mean ± s.e.m. (*n* = 288). Student's *t*-test was used to generate the *P* values. The presence of the same lowercase letter denotes a non-significant difference between means (*P* > 0.05)

studies have shown that the loss-of-function mutation of the
GS3 gene was associated with the formation of much larger
grains[15], whereas the dep1-1 allele was correlated with an
increased grain numbers[8,10]. However, the molecular
mechanisms of the two Gγ subunits-mediated improvement of
grain size and grain number remain unclear. In this study, we
have demonstrated that both GS3 and DEP1 interact directly
with transcription factor OsMADS1, and function as cofactors
to promote transcriptional activity of OsMADS1, and coop-
eratively regulate a common set of target genes, which in turn
determine grain size and shape. Our findings of the interaction
between Gβγ subunits and MADS-domain transcription fac-
tors reveal a new molecular framework for coupling floral
organ identity and growth with plasma-membrane-associated
G-protein complexes, which links intercellular responses and
environmental changes.

Haplotype analysis of the OsMADS1 locus revealed that the
lgy3 allele involving truncating splice-site mutation is common
within O. nivara accessions and tropical japonica germplasm, but
it does not appear to occur within the elite indica and temperate
japonica rice varieties (Supplementary Data 1). This result indi-
cates that the lgy3 allele has not been used to breed elite indica
and temperate japonica varieties. We have shown that the dep1-1
allele has been widely used in high-yielding japonica rice breeding
programs in China[8,10]. Therefore, we employed QTL pyramiding
based on combinations of the lgy3 and dep1-1 alleles with
molecular marker-assisted selection. Introduction of the lgy3
allele into the high-yielding japonica variety WYJ7 carrying the
dep1-1 allele resulted in a 10.4% advantage in grain yield. More
importantly, it also was associated with much better quality in
terms of grain length-to-width ratio and grain chalkiness (Fig. 2,
Supplementary Figs. 8, 9 and Supplementary Data 2). On the
other hand, the gs3 allele has also been widely used in indica
hybrid rice breeding programs[15,17]. In the Liangyoupeijiu[44]
(PA64S x 9311, a high-yielding F$_1$ hybrid rice cultivated in
Yangtze River area in China) genetic background, we found that
pyramiding of lgy3 and gs3 alleles enhanced overall grain yield (by
about 7%), with improved grain quality (Supplementary Figs. 8, 9
and Supplementary Data 2). Thus, combining lgy3 allele with
high-yield potential-associated dep1-1 and gs3 alleles provides a
new strategy in breeding simultaneously for higher grain yield
and better grain quality in rice above what is currently achievable.

## Methods

**Plant materials and growing conditions**. A set of 250 RILs population was bred
from the cross between L-204 and WYJ7. Details of the germplasm used for the
positional cloning and haplotype analysis have been described elsewhere[7,18]. The
NILs plants carrying contrasting combinations of the qLGY3 and qDEP1 loci,
qLGY3, qGS3 loci were bred by crossing RIL186 six times with the recurrent
parents Nipponbare, WYJ7, RD23, and 9311, respectively. Field-grown NIL plants
were raised under standard paddy conditions at two experimental stations, one
located in Lingshui (Hainan Province) and the other in Hefei (Anhui Province).

**The positional cloning of qlgy3**. Fine-scale mapping of qlgy3 was based on 4,247
BC$_2$F$_2$ populations and 653 BC$_3$F$_2$ populations derived from the backcross between
a selected RIL186 and an indica rice variety RD23 (RD23 as the recurrent parent).
The primer sequences used for map-based cloning and genotyping assays were
given in Supplementary Data 4.

**Transgene constructs**. The OsMADS1$^{lgy3}$ coding sequence was amplified from cv.
L-204, the OsMADS1 coding sequence and its promoter regions lying 1.9 Kbp
upstream of the transcription start site and 3′-UTR lying 1.5 Kbp downstream of
the termination site were amplified from cv. Nipponbare, and then inserted into
the pCAMBIA1300 (CAMBIA) to generate the pOsMADS1::OsMADS1 and pOs-
MADS1::OsMADS1$^{lgy3}$ expression cassettes. A 317 bp cDNA fragment of
OsMADS1 was amplified and used to construct both the pActin::RNAi-OsMADS1
and pOsMADS1::RNAi-OsMADS1 transgenes as described elsewhere[3]. To generate
the p35S::OsMADS1$^{lgy3}$-GFP and p35S::OsMADS1-GFP fusions, full-length cDNAs
of OsMADS1 and OsMADS1$^{lgy3}$ were amplified, and then cloned into p35S::GFP-
nos vector[4]. The transgenic rice plants were generated by Agrobacterium-mediated

transformation as described elsewhere[3]. Relevant primer sequences were given in
Supplementary Data 5.

**RNA isolation and quantitative PCR**. The total RNA was extracted using TRIzol
reagent (Invitrogen), and then treated with RNase-free DNase I (Invitrogen)
according to the manufacturer's protocol. The RNA was reverse-transcribed
using a cDNA synthesis kit (TRANSGEN). Quantitative real-time PCR (qRT-
PCR) was performed from cDNA using SYBR Green qPCR mix (TRANSGEN,
AQ101) following the manufacturer's instructions on an Applied Biosystems
7900HT Fast Real-Time PCR System. Fold changes were calculated using the ΔCt
method, each assay was replicated by at least three time with three biological
replicates (independent RNA preparations). The rice Actin3 (LOC_Os03g61970)
was used as a reference. Relevant primer sequences were given in Supplementary
Data 6.

**Yeast two-hybrid assays**. Yeast two-hybrid assays were performed as described
elsewhere. Full-length cDNAs of DEP1, dep1-1, OsMADS3, OsMADS5, OsMADS7,
OsMADS13, OsMADS14, OsMADS15, OsMADS16, and OsMADS55 were amplified
and then subcloned into pGBKT7 vector (Takara Bio Inc.), and full-length cDNAs
of OsMADS1 and OsMADS1$^{lgy3}$ were inserted into pGADT7 (Takara Bio Inc.). All
clones were validated by sequencing. The bait and prey vectors were co-
transformed into yeast strain AH109, and β-galactosidase assays were performed
according to the manufacturer's protocol (Takara Bio Inc.). The dep1-1 protein was
used as a bait to screen a cDNA library from poly(A)-containing RNA isolated
from young (0.2–6 cm length) panicles. Experimental procedures for screening and
plasmid isolation were performed following the manufacturer's instructions.
Relevant primer sequences were given in Supplementary Data 5.

**SFLC assays**. Full-length cDNAs of RGA1, RGB1, RGG1, RGG2, DEP1, dep1-1,
GS3, OsGGC2, OsMADS4, OsMADS13, OsMADS14, OsMADS34, OsMADS55,
OsMADS58, OsMADS1$^{lgy3}$, and deleted and non-deleted versions of either
OsMADS1 or DEP1 were amplified and the amplicons were inserted into
pCAMBIA1300-35S-Cluc-RBS or pCAMBIA1300-35S-HA-Nluc-RBS vectors[58] to
generate the required fusion transgenes. Two plasmid vectors for testing the
protein–protein interactions (such as DEP1-nLUC and OsMADS1-cLUC), toge-
ther with the p19 silencing plasmid, were co-transfected into tobacco (N. ben-
thamiana) leaves via Agrobacterium tuiefaciens infiltration, and LUC activity was
measured as described elsewhere[59]. The injected leaves were detached after 36–48 h
later and sprayed with 1 mM luciferin (Promega, E1605). LUC signal was captured
using a cooled CCD-imaging apparatus (Berthold, LB985). Each assay was repeated
at least three times. Relevant primer sequences were given in Supplementary
Data 5.

**Co-immunoprecipitation assays**. Full-length cDNAs of DEP1, dep1-1, and
OsMADS1 were amplified, and then inserted into either PUC-35S-HA-RBS or
PUC-35S-flag-RBS vector[31]. A. thaliana protoplasts were transfected with 100 μg of
plasmid and incubated overnight under low light intensity. Total protein was
extracted from harvested protoplasts by treating with 50 mM HEPES (pH7.5), 150
mM KCl, 1 mM EDTA, 0.5% Trition-X 100, 1 mM DTT, and proteinase inhibitor
cocktail (Roche LifeScience). The lysates were incubated with Flag M2 Affinity Gel
(Sigma-Aldrich, A2220) at 4 °C for at least 4 h, then rinsed 5–6 times in the
extraction buffer and eluted with 3× Flag peptide (Sigma-Aldrich, F4709). The
immunoprecipitates were electrophoretically separated by SDS-PAGE and trans-
ferred to a nitrocellulose membrane (GE Healthcare). Proteins were detected by
treating the membranes with anti-HA (1:5,000, MBL, M180-7) or anti-DDDDK-
tag mAb-HRP-DirectT antibodies (1:10,000, MBL, M185-7). Uncropped blots were
shown in Supplementary Fig. 10.

**ChIP-PCR assays**. ChIP assays were performed as described elsewhere[60]. 2 g rice
seedlings were grinded with liquid nitrogen and fixed with 1% formaldehyde under
vacuum. After nuclei were isolated and lysed, chromatin was ultrasonic fragmented
on ice to an average size of 500 bp. The supernatant was firstly blocked with protein
A agarose beads pre-absorbed with sheared salmon sperm DNA (Upstate, 16-157),
and then a small part of the buffer was setted aside to serve as input. Immuno-
precipitations were performed with anti-LGY3 antibody (1:3,000, ABclonal) at 4 °
C. At the same time, an equal volume of supernatant was prepared without any
antibody for mock sample. Finally, reverse-crosslink and precipitated DNA served
as the templates for qRT-PCR. The primer sequences were given in Supplementary
Data 6.

**EMSA assays**. The OsMADS1$^{lgy3}$ and OsMADS1 coding sequences were amplified
and inserted into pGEX-4T-1 vector (GE Healthcare). GST, GST-OsMADS1, and
GST-OsMADS1$^{lgy3}$ were purified using Glutathione Sepharose 4B (GE Healthcare,
17-0756) following the manufacturer's instructions. DNA probes were amplified
and labeled using a biotin label kit (Invitrogen). DNA gel-shift assays were per-
formed using a LightShift Chemiluminescent EMSA Kit (Thermo Fisher Scientific,
20148) as described elsewhere[18]. The relevant primer sequences were given in
Supplementary Data 6.

**Transactivation activity assays**. The ~2 Kbp DNA fragment of the promoter region of either OsMADS55, OsKANADI24, OsPIN1a, OsARF9, or OsARF14 was amplified from cv. Nipponbare, and used to generate reporter plasmids containing a specific promoter fused to LUC. Full-length cDNAs of OsMADS1, OsMADS1[gy3], and deleted versions of OsMADS1 were amplified and fused to GAL4BD, and then inserted into pUC19 vector[61] to generate effector plasmid vectors (e.g., pGALBD-OsMADS1). The transactivation analysis was performed using rice protoplasts as described elsewhere[18]. The rice protoplasts were transfected with different combinations of vectors overnight, then harvested and lysed for the detection of firefly luciferase activity (Promega, E1960), according to the manufacturer's recommendations. Relevant PCR primer sequences were given in Supplementary Data 5.

**Data availability**. The authors declare that all relevant data supporting the findings of this study are included in the main manuscript file or Supplementary Information or are available from the corresponding author upon reasonable request.

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

## Acknowledgements

We thank Prof. Nick Harberd for the critical comments on the manuscript. This research was supported by grants from National Natural Science Foundation of China (91635302 and 91335207), the National Key Research and Development Program of China (2016YFD0100401), the Chinese Academy of Sciences (XDA08010101), and the National Special Project of China (2014ZX0800935B).

## Author contributions

Q.L. and R.H. performed most of the experiments; R.H., J.Z., and D.T. conducted the QTL analysis; R.H., K.W., and S.W. constructed the NILs; R.H., Y.W., and Y.P. con-structed performed field experiments; J.Z., Q.L., and J.C. performed yeast two-hybrid screen; R.H., K.W., and Y.Y. characterized the phenotype of transgenic plants; K.W. and R.H. analyzed grain quality; and Q.L., X.X., and R.H. were responsible for haplotype analysis. X.F. designed the experiments and wrote the manuscript. All the authors have discussed the results and contributed to the drafting of the manuscript.

## Additional information

**Competing interests:** The authors declare no competing financial interests.

