## [Peer Review File · Nature Communications]

Reviewers' comments:

Reviewer #1 (Remarks to the Author):

In the manuscript "Heterotrimeric G proteins determine grain size through interaction with MADS domain transcription factors in rice" Liu et al. present data showing that the control of grain size and yield depends on the interaction between OsMADS1 and the heterotrimeric G- γ subunits.

Following a QTL mapping approach using 250 RILs, generated by a cross between the japonica high yielding variety WYJ7 dep1-1 and an American japonica variety, L-204, which generates long and slender seeds, the authors were able to identify a 8.8kB genomic region that only contained the CDS plus 3'UTR of MADS-box transcription factor OsMADS1. The identified L-204 allele showed a splice site located insertion-deletion polymorphism, inducing the truncation of the terminal 37 residues plus the addition of 5 residues at the C-terminal. Analysis of a near isogenic NPB-lgy3 line showed the formation of long seeds with regular cell elongation, indicating that the lgy3 allele promotes cell proliferation. Native and ectopic Expression of L-204 lgy3 cDNA induced the formation of long grains, whereas native expression of LGY3 cDNA caused no phenotypic alterations, indicating that product of OsMADS1lgy3 acts as a dominant negative regulator of OsMADS1 function. In addition transcriptional activity assays indicate that the presence of DEP1 enhances the activity and that the lgy3 allele shows reduced activity when compared to the full-length LGY3 allele, suggesting that DEP1-OsMADS1 interaction promotes the transactivation activity of OsMADS1.

Transcriptional profiling revealed 451 genes that were co-operatively regulated by the DEP1-OsMADS1 interaction, including other MADS-box genes and genes involved in Auxin signaling.

Subsequent Y2H assays showed that both DEP1 and dep1-1 proteins interacted with OsMADS1 and this interaction was further confirmed using Luciferase complementation assays and coIPs in planta. The truncated OsMADS1lgy3 version also interacted with DEP1 and dep1-1 proteins and analysis of various domain-based deletions of OsMADS1 indicated that both I and K domains were responsible for its interaction with DEP1. Finally the authors present a data set, showing that the combination of the grain size3 (gs3) and the OsMADS1lgy3 alleles enhanced yield by around 7%.

Our understanding of heterotrimeric G protein signaling in crops is still quite limited, therefore any contribution to this field is of great general interest, especially if its related to genetic and molecular mechanisms controlling yield. The data about the genetic identification of the

OsMADS1lgy3 allele and the phenotypic consequences of its interaction with the dep1-1 and gs3 alleles are robust and well presented, and represent clearly the strong part of the manuscript. The functional characterisation showing that OsMADS1 interacts with heterotrimeric G proteins, especially DEP1 and GS3, to control grain size are unfortunately not as convincing. While it is eminent that the OsMADS1lgy3 allele affects grain size, the observation that it does interact with all four tested γ -subunits questions the specificity and biological relevance of this postulated interaction, as it also implies that all four tested γ -subunits are nuclear localized. Published expression data, based on subcellular membrane fractionation, however indicate that RGG1 and RGG2 are localized to the plasma membrane (Kato et al., 2004). In the case of DEP1 the published data are controversial, Zhou et al. report that DEP1 is localized to the plasma membrane (2009), whereas Huang et al. report it DEP1 to be expressed in the nucleus (2009). I therefore believe, that the validation of DEP1 and GS3 as nuclear localized proteins is an essential experimental data set, that should to be included in this manuscript prior to acceptance. In this respect it is also a bit disappointing that it remains mechanistically unclear how the truncated OsMADS1lgy3 allele increase transcription of the identified target genes and how it relates to heterotrimeric G protein signaling.

Additional points:

The split Luciferase complementation assays should not be declared as BiFC

Citation 25 does not prove that the G $\beta\gamma$ dimer acts as a functional monomer.

The title is misleading as not all heterotrimeric G proteins are involved in the postulated OsMADS1 interaction.

The concluding statement: „Given the evolutionary conservation of both the G protein and 1 MADS-box gene families, the interaction between G $\beta\gamma$ subunits and MADS-domain proteins establishes a new molecular framework for the control of stem cell function and floral organ development under fluctuating environmental conditions.“ is also a bit far reaching as the analyzed phenotypic alterations within this manuscript do not relate to phenotypes associated with defects in meristem development or stem cell homeostasis.

The manuscript would also benefit from a more detailed introduction into the current understanding of heterotrimeric G protein signaling in plants. This aspect is also missing in the interpretation of the data, especially with respect to the underlying molecular mechanisms.

Some of the figures appear to be mislabeled:

Figure 1e: Labeling is identical

Figure 1f: Labeling is identical

Figure 2d: Labeling of delta K and delta I

Figure 3a: Labeling of NILs are identical

In summary

Kato, C., T. Mizutani, H. Tamaki, H. Kumagai, T. Kamiya, A. Hirobe, Y. Fujisawa, H. Kato, and Y. Iwasaki (2004). Characterization of heterotrimeric G protein complexes in rice plasma membrane. *Plant J.* 38(2): p. 320-31.

Zhou Y, Zhu J, Li Z, Yi C, Liu J, Zhang H, Tang S, Gu M, Liang G: Deletion in a quantitative trait gene qPE9-1 associated with panicle erectness improves plant architecture during rice domestication. *Genetics* 2009, 183:315-324.

Huang X, Qian Q, Liu Z, Sun H, He S, Luo D, Xia G, Chu C, Li J, Fu X: Natural variation at the DEP1 locus enhances grain yield in rice. *Nat Genet* 2009, 41:494-497

Reviewer #2 (Remarks to the Author):

The grain length is very important determinant of rice appearance grain quality and yield, which is a complex quantitative trait by numerous QTLs and less influencing by environment factors. There are many QTLs for grain shapes were cloned before. Dr. Fu's group successfully identified a new QTL, qLGY3, which control grain length and grain yield. It is encoded a transcript factor OsMADS1, this is the 1st time to find that QTL for rice grain is associated with the MADS domain in rice. And this paper demonstrates the interactions of OsMADS1 with DEP1 and GS3 could enhance the regulations of their common target genes thereby controlling grain size. Abundant field trails also show a potential for higher grain yield and better grain quality rice breeding.

I will focus my comments here on some conclusions that must be considered to make the paper a robust contribution.

1) Page 5 line 11: "At the same time, WYJ7-lgy3-dep1-1 plants produced much better quality in terms of grain length to width ratio and grain chalkiness (Supplementary Fig. 4b, 5b and Supplementary Table 2). Thus, pyramiding of lgy3 and dep1-1 alleles provides a promising strategy for simultaneously improving rice yield and grain quality."

...and also described in the other sections.

I find the transparency of lgy3 NILs plants, showed in the Supplemental Fig.4, is worse than LGY3 NILs plants, which is paradoxical taking the SEM images of Supplemental Fig.5 into account.

2) Fig. 1e, f: The notes under the figure are same.

3) Page 4 line 15, Supplemental Fig.3: “The mean length of NPB-lgy3 outer epidermal cells was indistinguishable from that of the equivalent cells in NPB-LGY3 (Supplementary Fig. 3e, f), indicating that the lgy3 allele enhanced longitudinal grain growth by promoting cell proliferation.”

Additional evidences should be provided to validate the hypothesis that enhanced longitudinal grain growth is associated with cell proliferation.

4) Page 4 line 26: “Thus the product of OsMADS1lgy3 appears to act as a dominant negative regulator of OsMADS1 function.”

Why “The expression of the Nipponbare LGY3 cDNA driven by its native promoter had no effect on grain length” and how many transgenic plants have been generated?

If there are dosage effect in controlling grain size?

Reviewer #3 (Remarks to the Author):

This manuscript is described that rice G gamma subunits are interacted with OsMADS1. It represents a major amount of work and largely contributes to understanding of plant G protein signaling. Reviewer wrote some comments in this manuscript. Reviewer thinks that there are some overstatements in this manuscript.

Almost all readers learned that the heterotrimeric G protein is mainly worked on the plasma membrane as general concept. Unlike the general concept, this work presents that rice G protein beta subunit and some gamma subunits possibly function in nucleus, not plasma membrane, because authors focus transcription activity and OsMADS1 is localized in nucleus. If there are some papers describing that the heterotrimeric G protein beta and gamma dimer is localized in nucleus and has some function to play in mammals and yeast, authors should cite preceding studies and emphasize generality.

In abstract, authors described that OsMADS1 is a direct target of the G protein beta- gamma dimer. OsMADS1 interacts with beta subunit or gamma subunit, respectively in this paper. However, reviewer feels that this sentence is overstatement, because it is not studied whether the purified beta-gamma dimer can interact with OsMADS1 or not.

In abstract, authors described that GS3 and DEP1 interact with OsMADS1. On the other hand, in Fig. 4 (a), four kinds of gamma subunits interact with OsMADS1. Is there some reason which the Ggamma1 and Ggamma2 are not written in abstract? As related to the Fig 4 (a), Gbeta is localized in plasma membrane and nucleus (Nature Genetics 2014), DEP1 is localized in plasma membrane and nucleus (Nature Genetics 2014 supplement Fig15), DEP1 is localized in plasma

membrane and nucleus in the presence of Gbeta (Nature Genetics 2014), OsMADS is localized in nucleus (supplement Fig. 2 in this study). As four kinds of gamma subunits are interacted with OsMADS1 (Fig.4 in this study), the interaction between all gamma subunits and OsMADS1 should be carried out in nucleus. Can you show or cite the data for localization of Ggamma1, Ggamma2 and GS3?

In Fig. 2 (c), authors studied the interaction between the various domain-based deletions of OsMADS1 vs DEP1 and found that K domain in OsMADS1 was important for interaction with DEP1. Reviewer hopes that authors also find out the specific amino acid sequences in DEP1, GS3, Ggamma1 and Ggamma2, which are necessary for interaction with OsMADS1. Reviewer thinks that it is important for identification of the specific amino acid sequence of gamma subunits necessary to interaction with Gbeta subunit or OsMADS1. If the target sequence of Ggamma subunits to Gbeta and OsMADS1 is same, G gamma subunits may move from Gbeta to OsMADS1.

The lgy3 mutant increases yield by enlarged seed. The dep1-1 mutant increase yield by increment of seed number, not seed size. Author should pick up genes commonly up-regulated in WYJ7-lgy3-DEP1 and WYJ7-LYG3-dep1-1, compared with WYJ7-LGY3-DEP1, if authors aim to find out the common set for yield, not seed size. As this reason, authors should add RNA-seq analysis data in WYJ7-lgy3-DEP1 in Fig.3 (b).

Reviewer feels that descriptions of DEP1 allele are sometimes confused. Reviewer described that OsMADS1 and DEP1 are repressors of grain size (p7, lane 6-7). DEP1 is consisted with the gamma domain plus cystein-rich domain. The dep1-1 mutant is consisted with only gamma domain. The dep1-1 mutant shows slightly shorten seed and set large number of seed. Reviewer agrees that the only gamma domain is repressor of grain size. As the dep1-32 mutant has 93 amino acid residues, it may have functional gamma domain. As these reason, reviewer does not know phenotypes of DEP null allele. Now, the function of full length DEP1, which is consisted with the gamma domain plus cystein-rich domain, may be hard to describe.

Question or Error

Fig.1 Plant names in (e) and (f)

Wrong: WYJ7-LGY3-dep1-1(left) WYJ7-LGY3-dep1-1(Right)

Correct: WYJ7-LGY3-dep1-1(left) WYJ7-lgy3-dep1-1(Right)

Fig.4 Plant names in (j), (k), (l)

Wrong: RD23-LGY3-GS3, RD23-LGY3-gs3, ,RD23-LGY3-gs3, RD23-lgy3-gs3

Correct: RD23-LGY3-GS3, RD23-lgy3-GS3, ,RD23-LGY3-gs3, RD23-lgy3-gs3

Response (R) to reviewer's question (Q)

The following contains a “point-by-point” response to the reviewers' comments and an indication of additional data and changes to the manuscript made in response to these comments:

Reviewer #1 (Remarks to the Author)

Q1: *While it is eminent that the $OsMADS1^{lgv3}$ allele affects grain size, the observation that it does interact with all four tested γ -subunits questions the specificity and biological relevance of this postulated interaction, as it also implies that all four tested γ -subunits are nuclear localized. Published expression data, based on subcellular membrane fractionation, however indicate that RGG1 and RGG2 are localized to the plasma membrane (Kato et al., 2004). In the case of DEP1 the published data are controversial, Zhou et al. report that DEP1 is localized to the plasma membrane (2009), whereas Huang et al. report it DEP1 to be expressed in the nucleus (2009). I therefore believe, that the validation of DEP1 and GS3 as nuclear localized proteins is an essential experimental data set that should to be included in this manuscript prior to acceptance.*

R1: We agree the reviewer's comments. In fact, we also observed that both DEP1-GFP fusion protein and mutated dep1-GFP fusion protein were localized to the plasma membrane in onion epidermal cells.

However, we found that *dep1*-GFP fusion protein was detectable in both the plasma membrane and the nucleus in the transgenic rice plants. These results have been shown in our previous papers (Huang *et al.*, Nature Genetics, 2009; Sun *et al.*, Nature Genetics 2014). In the further experiments, we generated the transgenic rice plants carrying either *pActin::HA-NES-dep1* or *pActin::HA-NLS-dep1* construct. As shown in the following figures, either nuclear localized HA-NLS-*dep1* fusion protein or cytosol-localized HA-NES-*dep1* fusion protein resulted in a semidwarf phenotype with a reduced size of grain, suggesting that both nuclear -localized and cytosol-localized *dep1* fusion proteins are required for G protein signalling.

We showed that the RGB1 (rice G protein β subunit)-GFP fusion protein was able to be detected in both the plasma membrane and the nucleus in the transgenic rice plants, and the up-regulation of *RGB1-GFP* was associated of the semidwarfism with a reduced grain size (Sun *et al.*, Nature Genetics, 2014). These results suggest that RGB1-*dep1* dimer was able to translocate into the nucleus, and consequently trigger the transcription of those target genes via the regulation of transcription factor function or unknown mechanisms in the nucleus. Consistent with the note that the G protein β subunit interacted tightly with the $G\gamma$ subunits as described in previous paper (Kato *et al.*, 2004), the GS3 protein interacts directly with the RGB1 protein. The implication is that the RGB1-GS3 dimer is able to translocate into the nucleus. As shown in the following figures, both cytosol-localized and nuclear-localized GFP-GS3 was also detectable in tobacco epidermal cells after infiltration with *Agrobacterium* suspension cultures, which was similar to that of GFP-RGB1 in tobacco leaves.

To investigate whether the GS3 protein is able to translocate into the nucleus *in planta*. We generated the transgenic rice plants carrying the *pActin::GS3-GFP* construct. As shown in the following figures, the transgenic rice plants overexpressing the fusion gene encoding GS3-GFP displayed a semidwarf phenotype with small grain, which was similar to that of the transgenic plants carrying the *pActin::myc-GS3* construct as previously described (Sun *et al.*, Nature Genetics, 2014).

More importantly, the GS3-GFP fusion protein was detectable in the nucleus of rice protoplasts isolated from leaf sheath of 10-day-old seedlings of the transgenic plants. Taken together, these results suggest that both DEP1 and GS3 are able to translocate into the nucleus. In the revised version, we have added additional experiments and data (**Supplementary Fig. 9**) and modified the text as suggested.

Q2: *In this respect it is also a bit disappointing that it remains mechanistically unclear how the truncated $OsMADS1^{lgy3}$ allele increase transcription of the identified target genes and how it relates to heterotrimeric G protein signaling.*

R2: According to genetic analysis (**Supplementary Fig. 1** and **Supplementary Fig. 3**), the $OsMADS1^{lgy3}$ allele functions as a dominant negative regulator of grain length and 1,000-grain weight. In the represent of the $OsMADS1^{lgy3}$ allele in the WYJ7 genetic background, the grain length of WYJ7- $lgy3$ - $dep1-1$ plants was longer than that of grain produced by WYJ7- $lgy3$ - $dep1-1$ plants. Moreover, the DEP1 protein interacted with either OsMADS1 or $OsMADS1^{lgy3}$ protein, and further RNA-seq and ChIP experiments demonstrated that DEP1 acts as a functional OsMADS1 cofactor in controlling grain size and shape through the regulation of the common target genes, such as $OsMADS55$ and $OsARF9$. Next, we showed that conserved K-domain is required for the interaction between DEP1 and OsMADS1 proteins. Lacking of C-domain of OsMADS1 (i.e. the truncated $OsMADS1^{lgy3}$ protein) was its weak transcription activity of the target gene, while the lack of C-domain did not interfere with DNA binding activity of OsMADS1 and interactions with either DEP1 or $dep1$ protein.

Further transcriptional activity assays showed that transactivation activity of OsMADS1 was less than that of either truncated $OsMADS1^{\Delta K}$ (lacking K domain) or truncated $OsMADS1^{\Delta IK}$ (lacking both I and K domains), suggesting that C domain-dependent transactivation activity of OsMADS1 was influenced by the present of I and K domains. Interestingly, transactivation activity of OsMADS1 was substantially promoted when both OsMADS1 and DEP1 were combined in transient activity assays, whereas its activity was unaffected when $OsMADS1^{\Delta K}$ (or $OsMADS1^{\Delta IK}$) and DEP1 were combined. These results suggested that K domain could repress C-domain dependent transactivation activity of OsMADS1, and DEP1 promotes the transactivation activity of OsMADS1 through its interaction with K domain. Although deletion of C domain of OsMADS1 did not affect the interaction between $OsMADS1^{lgy3}$ and DEP1, $OsMADS1^{lgy3}$ did not show transcription factor activity, and the interaction between DEP1 and K domain of $OsMADS1^{lgy3}$ could not promote the transactivation activity of

OsMADS1^{lg γ 3}. Thus, DEP1 regulates grain size via its activation of OsMADS1, and DEP1 acts as cofactors to enhance OsMADS1 transcriptional activity and promote the co-operative transactivation of common target genes. In the revised version, we have modified the text to more accurately describe the role of DEP1 (G protein signaling) in the regulation of grain size through the DEP1-OsMADS1 interaction.

Q3: *The split Luciferase complementation assays should not be declared as BiFC*

R3: We have corrected it as suggested.

Q4: *Citation 25 does not prove that the G $\beta\gamma$ dimer acts as a functional monomer.*

R4: We have corrected it as suggested.

Q5: *The title is misleading as not all heterotrimeric G proteins are involved in the postulated OsMADS1 interaction.*

R5: We agree the reviewer's comments. In the revised version, we have changed the title as suggested. The new title is: **G-protein $\beta\gamma$ subunits determine grain size through interaction with MADS-domain transcription factors in rice.**

Q6: *Given the evolutionary conservation of both the G protein and MADS-box gene families, the interaction between G $\beta\gamma$ subunits and MADS-domain proteins establishes a new molecular framework for the control of stem cell function and floral organ development under fluctuating environmental conditions. " is also a bit far reaching as the analyzed phenotypic alterations within this manuscript do not relate to phenotypes associated with defects in meristem development or stem cell homeostasis.*

R6: We agree reviewer's comments. In the revised version, we have modified the text as suggested.

Q7: *The manuscript would also benefit from a more detailed introduction into the current understanding of heterotrimeric G protein signaling in plants. This aspect is also missing in the interpretation of the data, especially with respect to the underlying molecular mechanisms.*

R7: Thanks for the reviewer's insightful comments. In the revised version, we have modified the text as suggested.

Q8: *Some of the figures appear to be mislabeled:*

Figure 1e: Labeling is identical

Figure 1f: Labeling is identical

Figure 2d: Labeling of delta K and delta I

Figure 3a: Labeling of NILs are identical

R8: We have corrected it as suggested.

Q9: *Kato, C., T. Mizutani, H. Tamaki, H. Kumagai, T. Kamiya, A. Hirobe, Y. Fujisawa, H. Kato, and Y. Iwasaki (2004). Characterization of heterotrimeric G protein complexes in rice plasma membrane. Plant J. 38(2): p. 320-31.*

Zhou Y, Zhu J, Li Z, Yi C, Liu J, Zhang H, Tang S, Gu M, Liang G: Deletion in a quantitative trait gene qPE9-1 associated with panicle erectness improves plant architecture during rice domestication. Genetics 2009, 183:315-324.

Huang X, Qian Q, Liu Z, Sun H, He S, Luo D, Xia G, Chu C, Li J, Fu X: Natural variation at the DEPI locus enhances grain yield in rice. Nat Genet 2009, 41:494-497

R9: In the revised version, we have added these references as suggested.

Reviewer #2 (Remarks to the Author)

Q1: “At the same time, WYJ7-*lgy3-dep1-1* plants produced much better quality in terms of grain length to width ratio and grain chalkiness (Supplementary Fig. 4b, 5b and Supplementary Table 2). Thus, pyramiding of *lgy3* and *dep1-1* alleles provides a promising strategy for simultaneously improving rice yield and grain quality.” and also described in the other sections. I find the transparency of *lgy3* NILs plants, showed in the Supplemental Fig.4, is worse than *LGY3* NILs plants, which is paradoxical taking the SEM images of Supplemental Fig.5 into account.

R1: As shown in **Supplementary Figure 4**, the NIL plants carrying the *lgy3* allele exhibited a decreased percentage of chalky kernel and chalkiness degree when compared with the NIL plants carrying the *LGY3* allele in the same genetic backgrounds (i.e. *japonica* rice variety WYJ7 and *indica* rice variety RD23), which is consistent with the results shown in **Supplementary Table 2**. Thus, the *OsMADS1*^{*lgy3*} allele is associated with better quality of rice grain. Thanks for the reviewer’s insightful comments, we have modified **Supplementary Figure 4** and **Supplementary Figure 5** as suggested.

Q2: Fig. 1e, f: The notes under the figure are same.

R2: Yes, we made mistakes. We have corrected it as suggested.

Q3: “The mean length of NPB-*lgy3* outer epidermal cells was indistinguishable from that of the equivalent cells in NPB-*LGY3* (Supplementary Fig. 3e, f), indicating that the *lgy3* allele enhanced longitudinal grain growth by promoting cell proliferation.” Additional evidences should be provided to validate the hypothesis that enhanced longitudinal grain growth is associated with cell proliferation.

R3: The effects of allelic variation at the *OsMADS1* locus on grain size and shape were quantified in a field trial of NPB-*LGY3* and NPB-*lgy3* in the Nipponbare genetic background. Two NILs did not differ from one another with respect with grain width. The length of the NPB-*lgy3* grain was greater than those of

NIL-*LGY3* grain, resulting in the formation of a long and slender grain (**Supplementary Fig. 3b**). The down-regulation of the *OsMADS1* gene was correlated with the substantial increase in length of grain. The increased *OsMADS1* transcript abundance in transgenic NPB-*LGY3* plants harboring the *LGY3* allele of *OsMADS1* under the control of its native promoter did not alter rice grain length. In contrast, transgenic NPB-*LGY3* plants expressing the *OsMADS1*^{*lgy3*} allele formed longer grains than those formed by either NPB-*LGY3* or transgenic NPB-*LGY3* plants expressing *OSMADS1*^{*LGY3*} allele under the control of its native promoter or rice *Actin* promoter (**Supplementary Fig. 3g, h**). The inference was that *OsMADS1*^{*lgy3*} is a dominant negative allele, which regulates grain length.

There was little difference in cell length of outer epidermal cells in either the palea or the lemma (**Supplementary Fig. 3e, f**). These observations suggested that the *lgy3* allele might promote longitudinal growth by increasing cell proliferation. In addition, an inspection of longitudinal palea and lemma sections showed that cell length in the NPB-*lgy3* inner parenchyma cell layer was similar to that in the NIL-*LGY3* equivalent (data not shown). These results suggest that *OsMADS1*^{*lgy3*} is a dominant-negative regulator of grain length through promoting longitudinal cell proliferation. In revised version, we have modified the text as suggested.

Q4: *“Thus the product of *OsMADS1*^{*lgy3*} appears to act as a dominant negative regulator of *OsMADS1* function.” Why “The expression of the Nipponbare *LGY3* cDNA driven by its native promoter had no effect on grain length” and how many transgenic plants have been generated? If there are dosage effect in controlling grain size?*

R4: Previous studies have shown that constitutive expression of *OsMADS1*^{*LGY3*} cDNA resulted in abnormalities in grain development. In contrast, the transgenic rice plants expressing the *OsMADS1*^{*lgy3*} cDNA driven either by its native promoter or by the rice *Actin* promoter did not displayed abnormal grain development, but grains formed by these transgenic lines were longer than those formed by non-transgenic control plants (**Supplementary Fig. 3g, h**). In our experiments, over 10 independent transgenic plants carrying the *pOsMADS1::OsMADS1*

construct have been obtained, and we did not find substantial changes in grain length when compared with non-transgenic Nipponbare plants. In the revised version, we have modified the text as suggested.

Reviewer #3 (Remarks to the Author)

Q1: *Almost all readers learned that the heterotrimeric G protein is mainly worked on the plasma membrane as general concept. Unlike the general concept, this work presents that rice G protein beta subunit and some gamma subunits possibly function in nucleus, not plasma membrane, because authors focus transcription activity and OsMADS1 is localized in nucleus. If there are some papers describing that the heterotrimeric G protein beta and gamma dimer is localized in nucleus and has some function to play in mammals and yeast, authors should cite preceding studies and emphasize generality.*

R1: Thanks for the reviewer's insightful comments. In mammalian cells, following stimulation of G protein-coupled receptors (GPCRs) at the cell surface, the heterotrimeric G proteins are activated, and consequently detached G $\beta\gamma$ subunits regulate many classical effectors associated with the plasma membrane, including distinct phospholipase C β and voltage-gated calcium channels. Importantly, current studies suggested nuclear localization or translocation of the G $\beta\gamma$ subunit. For example, G $\beta\gamma$ co-localized with the AP-1 complex in the nucleus and recruited histone deacetylases (Chang, C.W. *et al.* Acute β -adrenergic activation triggers nuclear import of histone deacetylase 5 and delays G(q)-induced transcriptional activation. *J Biol Chem.* **288**, 192-204 (2013)) to inhibit AP-1 transcriptional activity (Robitaille, M. *et al.* G $\beta\gamma$ is a negative regulator of AP-1 mediated transcription. *Cell Signal* **22**, 1254-1266 (2010)). In the revised version, we have added additional references and modified the text as suggested.

Q2: *In abstract, authors described that OsMADS1 is a direct target of the G protein beta- gamma dimer. OsMADS1 interacts with beta subunit or gamma subunit, respectively in this paper. However, reviewer feels that this sentence is overstatement, because it is not studied whether the purified beta-gamma dimer can interact with OsMADS1 or not.*

R2: We agree the reviewer's comments, the G $\beta\gamma$ -OsMADS1 interaction was only confirmed by split firefly luciferase complementation and co-immunoprecipitation assays. In fact, we have tried so many times to express recombinant DEP1-His (GST-DEP1, BMP-DEP1, dep1-His, and GST-dep1 etc.) fusion proteins in *E.coli* bacteria and *Pichia pastoris* yeast. Unfortunately, we did not obtain soluble fusion proteins, and we could not perform the experiments to investigate whether the purified G $\beta\gamma$ dimer can interact with OsMADS1 or not. In the revised version, we have modified the abstract and the text as suggested.

Q3: *In abstract, authors described that GS3 and DEP1 interact with OsMADS1. On the other hand, in Fig. 4 (a), four kinds of gamma subunits interact with OsMADS1. Is there some reason which the Ggamma1 and Ggamma2 are not written in abstract? As related to the Fig 4 (a), Gbeta is localized in plasma membrane and nucleus (Nature Genetics 2014), DEP1 is localized in plasma membrane and nucleus (Nature Genetics 2014 supplement Fig15), DEP1 is localized in plasma membrane and nucleus in the presence of Gbeta (Nature Genetics 2014), OsMADS is localized in nucleus (supplement Fig. 2 in this study). As four kinds of gamma subunits are interacted with OsMADS1 (Fig.4 in this study), the interaction between all gamma subunits and OsMADS1 should be carried out in nucleus. Can you show or cite the data for localization of Ggamma1, Ggamma2 and GS3?*

R3: As mentioned above, recent data suggested nuclear localization or translocation of the G $\beta\gamma$ subunit in mammalian cells. For example, G $\beta\gamma$ co-localized with the AP-1 complex in the nucleus and recruited histone deacetylases to inhibit AP-1 transcriptional activity (Robitaille, M. et al., Cell Signal., 2010, 22:1254-1266; Chang, C.W. et al. J Biol Chem., 2013, 288:192-204). Previous studies have shown that both DEP1-GFP and RGB1-GFP fusion proteins localized in both the

plasma membrane and the nucleus in the transgenic rice plants, consistent with the note that the G β subunit interacted tightly with the γ subunits (Kato et al., Plant J., 2004, 38: 320-331). Overexpression of rice G β subunit (RGB1) and G γ subunits (i.e. RGG1, RGG2, GS3, and dep1) resulted in a semidwarf phenotype with a reduced grain size. Moreover, We found that the GS3-GFP fusion protein was detectable in the plasma membrane and the nucleus of protoplasts isolated from leaf sheath of 10-day-old seedlings of the transgenic rice plants carrying the *pActin::GS3-GFP* construct (**Supplementary Fig. 9**). These results suggest that rice G $\beta\gamma$ dimers can translocated into the nucleus, and may trigger transcription of the target genes via the regulation of transcription factor, which in turn determine grain size. Here, we demonstrated that transcription factor OsMADS1 interacted directly with rice G β subunit and four forms of G γ subunits, and OsMADS1 acts as a key downstream effector of G protein $\beta\gamma$ dimers. Because of the limitation of space in the abstract, we focus on two G protein γ subunits GS3 and DEP1, which regulate grain size and yield potential of rice. In the revised version, we have modified the abstract and text as suggested.

Q4: *In Fig. 2 (c), authors studied the interaction between the various domain-based deletions of OsMADS1 vs DEP1 and found that K domain in OsMADS1 was important for interaction with DEP1. Reviewer hopes that authors also find out the specific amino acid sequences in DEP1, GS3, Ggamma1 and Ggamma2, which are necessary for interaction with OsMADS1. Reviewer thinks that it is important for identification of the specific amino acid sequence of gamma subunits necessary to interaction with Gbeta subunit or OsMADS1. If the target sequence of Ggamma subunits to Gbeta and OsMADS1 is same, G gamma subunits may move from Gbeta to OsMADS1.*

R4: Thanks for the reviewer's insightful comments. As mentioned above, we focus on two G γ subunits GS3 and DEP1 because previous studies have shown their role in regulating grain size and grain yield in rice. Here, we demonstrated that DEP1 interacted directly with the K domain of OsMADS1 protein using split luciferase complementation and co-immunoprecipitation assays. We previously reported that the DEP1 protein comprises the GGL (G gamma like) domain, vWFC (von Willebrand factor type C) domain and TNFR (tumor necrosis factor

receptor)/NGFR (nerve growth factor receptor) domain (Sun et al., 2014, Nature Genetics). We have shown that the GGL domain of DEP1 is required for its interaction with RGB1. In the further experiments, we constructed various domain-based deletions of DEP1 for the split firefly luciferase complementation assays. As shown in the following figures, vWFC domain of DEP1 is required for the DEP1-OsMADS1 interaction. In the revised version, we have added additional experiments (**Supplementary Fig. 7**) and modified the text as suggested, so that it was more clearly described that the GGL domain of DEP1 is necessary for its interaction with G β subunit, and vWFC domain is necessary for its interaction with OsMADS1.

Q5: *The lgy3 mutant increases yield by enlarged seed. The dep1-1 mutant increase yield by increment of seed number, not seed size. Author should pick up genes commonly up-regulated in WYJ7-lyg3-DEP1 and WYJ7-LYG3-dep1-1, compared with WYJ7-LGY3-DEP1, if authors aim to find out the common set for yield, not seed size. As this reason, authors should add RNA-seq analysis data in WYJ7-lgy3-DEP1 in Fig.3 (b).*

R5: Thanks for the reviewer's insightful comments. As mentioned above, the *dep1-1* allele caused the increases in grain yield by the improvement of grain numbers per panicle. Actually, the rice plants carrying the *dep1-1* allele produced more, but smaller grains than its near-isogenic line, suggesting that *dep1-1* is a negative

regulator of grain size, but is a positive regulator of grain numbers per panicle. To identify common target genes for the *DEP1-OsMADS1* regulatory module, we have compared the genome-wide transcriptional profiles of developing panicles in the four NILs WYJ7-LGY3-*DEP1*, WYJ7-LGY3-*dep1-1*, WYJ7-*lgy3-DEP1* and WYJ7-*lgy3-dep1-1* using RNA-seq. As shown in **Fig. 1j**, there was little difference in grain numbers per panicle, but grain length of WYJ7-*lgy3-dep1-1* were longer than their equivalents in WYJ7-LGY3-*dep1-1*, suggesting that the genetic interaction between *lgy3* and *dep1-1* alleles regulates grain size, but not grain number. In this manuscript, we therefore focus on understanding of the molecular mechanisms underlying the OsMADS1-DEP1 interaction in regulating grain size, not grain number. In the revised version, we have modified the text as suggested.

Q6: *Reviewer feels that descriptions of DEP1 allele are sometimes confused. Reviewer described that OsMADS1 and DEP1 are repressors of grain size (p7, lane 6-7). DEP1 is consisted with the gamma domain plus cystein-rich domain. The dep1-1 mutant is consisted with only gamma domain. The dep1-1 mutant shows slightly shorten seed and set large number of seed. Reviewer agrees that the only gamma domain is repressor of grain size. As the dep1-32 mutant has 93 amino acid residues, it may have functional gamma domain. As these reason, reviewer does not know phenotypes of DEP null allele. Now, the function of full length DEP1, which is consisted with the gamma domain plus cystein-rich domain, may be hard to describe.*

R6: We agree the reviewer's comments. Indeed, we found the different *DEP1* alleles conferred different nitrogen response (Sun *et al.*, Nature Genetics, 2014), different number of grains per panicle, and different size of grains. For example, the rice plants carrying the *dep1-1* allele exhibits a reduced size of grains, but sets large number of seed. However, the rice plants carrying the *dn1-1* allele shows small grain and slightly increased number of grains per panicle although the truncated *dn1-1* protein contains the GGL domain and vWFC1 domain (Taguchi-Shiobara, *et al.* Breed. Sci., 2011, **61**:17-25). Because the *dep1-32* allele could encode 93 amino acid residues, it might have functional GGL domain. To produce the *DEP1* null mutant, we generated knockout mutants in

the WYJ7 genetic background using CRISPR/Cas9. Surprisingly, the phenotypes of the newly developed *dep1-C13* mutant, which has 13 amino acid residues and truncated *dep1-C13* proteins did not contain the GGL domain, were similar to that of *dep1-32* plant with respect to number of grains per panicle and grain size (data not shown). We found that the GGL domain could interact directly with the vWFC domain. However, it is unclear how the GGL and vWFC domains are successfully assembled into specific complexes in response to cues emanating from the internal plant's developmental status and/or from the external environment. Whatever the mechanism, our results indicate that the different *DEP1* alleles (or different domains) play different roles in the regulation of grain development from meristem inception to organ differentiation and meristem termination, and late growth and size of developing grains. In the revised version, we have modified the text as suggested.

Q7: *Question or Error*

Fig.1 Plant names in (e) and (f)

Wrong: WYJ7-LGY3-dep1-1(left) WYJ7-LGY3-dep1-1(Right)

Correct: WYJ7-LGY3-dep1-1(left) WYJ7-lgy3-dep1-1(Right)

Fig.4 Plant names in (j), (k), (l)

Wrong: RD23-LGY3-GS3, RD23-LGY3-gs3, ,RD23-LGY3-gs3, RD23-lgy3-gs3

Correct: RD23-LGY3-GS3, RD23-lgy3-GS3, ,RD23-LGY3-gs3, RD23-lgy3-gs3

R7: In the revised version, we have corrected it as suggested.

Reviewers' Comments:

Reviewer #1 (Remarks to the Author):

In the revised version of the manuscript, the authors have addressed most of my comments to my satisfaction. However, the additional data, supporting the notion that DEP1 and GS3 are nuclear localized are still not convincing.

In my understanding, it is very difficult to interpret localization results using over-expression systems, unless complementation studies indicate the biological functionality of the constructs. Ideally one would generate translational fusions within its genomic context, including the endogenous promoter, all introns, and exons plus the 3'UTR. Being aware that this is very time and cost intensive one possible experimental alternative could be the transient co-expression of RGB1 and DEP1-GFP or GS3-GFP in rice leaves, using particle bombardment and moderately expressing promoters since the native expression of DEP1 and GS3 in leaves might be too low to detect.

However, I also have concerns about the interpretation of the phenotypic *dep1* overexpression data using the actin promoter, as one should keep in mind that overexpression of one γ -subunit sequesters most likely all RGB1, thereby "inactivating" all other γ -subunits, which ultimately makes the interpretation of the phenotype very questionable. And indeed similar phenotypes, with reduced plant height and seed size, have been described for ectopic expression of RGG1 and RGG2.

Finally, I would also comment on the presented luciferase interaction assays. In case of the γ -subunits, it is important to remember that they are part of an obligatory dimer with RGB1, so the functional protein localization, especially when transiently expressed in a heterologous system, is only valid when both proteins are co-expressed. As far as I can tell, this has not been considered in the experimental setup of the luciferase complementation assays presented

Reviewer #2 (Remarks to the Author):

The authors had answered my questions and my suggestion is this manuscript should publish on Nature Communications.

Reviewer #3 (Remarks to the Author):

The authors have addressed my comments and have improved the paper. Reviewer thinks that this paper is an important contribution to the field of the G protein signaling in plants.

P10, lane 2-3:

“ as a functional monomer”

“monomer” should be changed another word.

P10, lane 29-30:

“indicating that the $G\beta\gamma$ dimers control grain size and shape via its regulation of OsMADS1.”

Reviewer thinks that this sentence is overstatement, because $G\gamma 1$ and $G\gamma 2$ do not have vWFC domain.

P26, lane 16, Figure legends, Figure 1, mislabel:

“WYJ7-gly3-dep1-1 plant”

Response (R) to reviewer's question (Q)

The following contains a “point-by-point” response to editor and reviewers' comments and an indication of additional data and changes to the revised manuscript made in response to these comments:

Reviewer #1

Q1: *In my understanding, it is very difficult to interpret localization results using over-expression systems, unless complementation studies indicate the biological functionality of the constructs. Ideally one would generate translational fusions within its genomic context, including the endogenous promoter, all introns, and exons plus the 3'UTR. Being aware that this is very time and cost intensive one possible experimental alternative could be the transient co-expression of RGB1 and DEPI-GFP or GS3-GFP in rice leaves, using particle bombardment and moderately expressing promoters since the native expression of DEPI and GS3 in leaves might be too low to detect.*

R1: We agree both editor and reviewer #1's comments. According to reviewer #1's suggestions, we performed suggested experiments and re-analyzed the localization of the GS3-GFP and DEPI-GFP fusion proteins using transient expression assays. Just as predicted, we did find the nuclear localization of GFP fusion proteins in rice protoplasts expressing either *GS3-GFP* or *DEPI-GFP* under the control of its native promoter. In addition, we generated transgenic rice plants expressing the *RGB1-GFP* fusion gene under the control of its native promoter. The RGB1-GFP fusion proteins were datable in the plasma membrane and the nucleus of root cells. In the revised version, we have added additional experiments and data (new **Supplementary Fig. 9**), and modified the text as suggested.

Q2: *However, I also have concerns about the interpretation of the phenotypic *depl* overexpression data using the actin promoter, as one should keep in mind that overexpression of one γ -subunit sequesters most likely all RGB1, thereby “inactivating” all other γ -subunits, which ultimately makes the interpretation of the phenotype very questionable. And indeed similar phenotypes, with reduced plant height and seed size, have been described for ectopic expression of RGG1 and RGG2.*

Editor's comments: *We also share the concerns of Reviewer #1, regarding the interpretation of DEPI overexpression data (which appears to give the same phenotype regardless of whether a NLS or NES sequence is used). In light of this, we feel that further evidence that DEPI acts in the nucleus as proposed, would strengthen the case for further consideration.*

R2: Thanks for the editor and reviewer's insightful comments. In mammalian cells, a lot of studies have shown that the G β subunit is a functional heterodimer that

forms a stable structural unit. G $\beta\gamma$ dimer association with G α prevents G $\beta\gamma$ from activating its effectors, following dissociation of G α , G $\beta\gamma$ is free to activate a large number of its own effectors. However, the mechanism of G $\beta\gamma$ interaction with its effectors is not entirely clear. Our previous studies revealed that either DEP1-GFP or RGB1-GFP fusion protein was able to be detected in the cytoplasm and the nucleus. The implication of this is that there may also be novel targets of rice G $\beta\gamma$ regulation in the nucleus.

We have shown that the NIL-*qNGR9/DEP1* plants carrying the *pDEP1::dep1-1* construct exhibited the semidwarf phenotype with the dense panicle architecture and reduced size of grains (*Supplementary Figure 2; Sun et al. Nature Genetics, 2014*). The result demonstrated that the expression of *dep1-1* under the control of its native promoter caused the reduced plant height and seed size, which were similar to that caused by the overexpression of *dep1-1* under control of the constitutive promoters (e.g., CaMV 35S promoter and rice *Actin* promoter). We next explored the relationship between the nuclear-localized *dep1-1* protein and its function in regulating plant height and grain size. We showed that the transgenic plants carrying the *pActin::HA-NLS-dep1-1* construct also displayed a semidwarf phenotype with the reduced grain size, which were slimmer to phenotypes produced by transgenic plants expressing *dep1-1* under control of either its native promoter and rice *Actin* promoter. These results indicated that the nuclear-localized *dep1-1* protein is associated with the formation of the semidwarfism and reduced size of grains in rice. Further demonstration of the interaction between plant-specific G γ subunits (GS3 and DEP1) and MADS1 transcription factor reveals a novel role of the nuclear-localized G $\beta\gamma$ subunits in the regulation of grain size. In this manuscript, we will focus on the function of the nuclear-localized G $\beta\gamma$ subunits, and remove the data of the transgenic plants carrying the *pActin::HA-NES-dep1-1* construct. In the revised version, we have added additional experiments and data (new **Supplementary Fig. 9**), and modified the text as suggested.

Q3: *I would also comment on the presented luciferase interaction assays. In case of the γ -subunits, it is important to remember that they are part of an obligatory dimer with RGB1, so the functional protein localization, especially when transiently expressed in a heterologous system, is only valid when both proteins are co-expressed. As far as I can tell, this has not been considered in the experimental setup of the luciferase complementation assays presented.*

R3: We agree reviewer's comments. G $\beta\gamma$ subunits act as a functional dimer to regulate a large number of its own effectors. The rice plants carrying the *dep1-1* allele exhibited reduced plant height and grain size, but mutation of *LGY3/OsMADS1* could restore the grain phenotypes. The two NILs plants, WYJ7-*lgy3-dep1-1* and WYJ7-*LGY3-dep1-1*, did not differ from one another with respect to plant height, but WYJ7-*lgy3-dep1-1* plants produced bigger grains than those formed by WYJ7-*LGY3-dep1-1* plants. These findings reveals a genetic interaction between

DEP1 and *OsMADS1* plays an important role in the regulation of grain size.

To investigate the molecular mechanisms of transcription factor OsMADS1 involved in G-protein signalling, we then performed a yeast two-hybrid screen. Surprisingly, we found that the LGY3 proteins interacted with G γ subunit DEP1. We also found the interaction between LGY3 and G γ subunit GS3 in yeast two-hybrid assays (data not shown). Further co-immunoprecipitation assays demonstrated that the DEP1 protein interacts directly with transcription factor OsMADS1 (**Figure 2**). It is known that, following dissociation of G α subunit, G $\beta\gamma$ subunits act as a functional dimer to regulate a large number of its own effectors. Interestingly, the split firefly luciferase complementation assays using tobacco leaf epidermal cells showed that the rice G β subunit (RGB1) interacts with the LGY3 protein. Co-immunoprecipitation assays also demonstrated that the LGY3-RGB1 interaction. Thus, either G β or G γ of G $\beta\gamma$ dimer interacts with LGY3 transcription factor. In addition, the up-regulation of *RGB1* was associated with the formation of reduced plant height (*Sun et al. Nature Genetics, 2014*) and reduced size of grains in the transgenic rice plants (**Figure 4**), whereas the mutation of the *LGY3* gene could restore the grain phenotypes caused by the increased abundance of RGB1. Taken together, our findings revealed that LGY3 acts as an effector of G $\beta\gamma$ dimer. In the revised version, we have modified the text as suggested.

Reviewer #3

Q1: P10, lane 2-3,

“ as a functional monomer”, monomer” should be changed another word.

R1: We have changed it as suggested.

Q2: P10, lane 29-30: *“indicating that the G $\beta\gamma$ dimers control grain size and shape via its regulation of OsMADS1.” Reviewer thinks that this sentence is overstatement, because G γ 1 and G γ 2 do not have vWFC domain.*

R2: We agree the reviewer’s comments. In the revised version, we have modified the text as suggested.

Q3: P26, lane 16, Figure legends, Figure 1, mislabel: *“WYJ7-gly3-dep1-1 plant”*

R3: We have corrected it as suggested.

Reviewers' Comments:

Reviewer #1 suggests accepting this manuscript in Remarks to Editor section.